# *Arabidopsis* SBT5.2 and SBT1.7 subtilases mediate C-terminal cleavage of flg22 epitope from bacterial flagellin

Sayaka Matsui [1], Saki Noda[1], Keiko Kuwata[2], Mika Nomoto [3], Yasuomi Tada [3], Hidefumi Shinohara [1,4] & Yoshikatsu Matsubayashi [1]✉

Plants initiate specific defense responses by recognizing conserved epitope peptides within the flagellin proteins derived from bacteria. Proteolytic cleavage of epitope peptides from flagellin by plant apoplastic proteases is thought to be crucial for the perception of the epitope by the plant receptor. However, the identity of the plant proteases involved in this process remains unknown. Here, we establish an efficient identification system for the target proteases in *Arabidopsis* apoplastic fluid; the method employs native two-dimensional electrophoresis followed by an in-gel proteolytic assay using a fluorescence-quenching peptide substrate. We designed a substrate to specifically detect proteolytic activity at the C-terminus of the flg22 epitope in flagellin and identified two plant subtilases, SBT5.2 and SBT1.7, as specific proteases responsible for the C-terminal cleavage of flg22. In the apoplastic fluid of *Arabidopsis* mutant plants deficient in these two proteases, we observe a decrease in the C-terminal cleavage of the flg22 domain from flagellin, leading to a decrease in the efficiency of flg22 epitope liberation. Consequently, defensive reactive oxygen species (ROS) production is delayed in *sbt5.2 sbt1.7* double-mutant leaf disks compared to wild type following flagellin exposure.

Plants defend themselves against pathogenic infection by using innate immune systems that evolved to recognize the presence of pathogens. This immune response process begins with the perception of molecular signatures (termed pathogen- or microbe-associated molecular patterns; PAMPs or MAMPs) that are shared among whole classes of microbes; sensing of MAMPs by pattern-recognition receptors triggers a basal resistance called pattern-triggered immunity (PTI)[1]. PTI responses include the production of reactive oxygen species (ROS), activation of the mitogen-activated protein kinase cascade, defense gene expression, and callose deposition[1].

One of the most extensively studied examples of MAMPs is flagellin, the major structural protein of the bacterial flagellum and a molecule that is distributed widely among distant bacterial species. Plants recognize flagellin monomers rather than polymerized flagellin. Notably, flg22, a 22-amino acid epitope peptide corresponding to the most conserved domain near the flagellin N-terminus, acts as a potent inducer of PTI in various plant species[2]. In *Arabidopsis*, the Leu-rich repeat transmembrane receptor kinase FLAGELLIN SENSING 2 (FLS2) directly recognizes flg22, leading to the initiation of immune signaling[3–6].

The flg22-FLS2 pair has been used as a model system to study PTI triggered by flagellin perception. In nature, however, flagellin is polymerized to form the flagella, resulting in the flg22 epitope being buried in the flagellin polymer structure[7]. Plants are thought to recognize flagellin monomers that are released as a result of leaks or as spillover during the construction of the flagella[8]. Although intact flagellin monomer protein shows some interaction with FLS2, the binding

[1]Division of Biological Science, Graduate School of Science, Nagoya University, Chikusa, Nagoya 464-8602, Japan. [2]Institute of Transformative Bio-Molecules, Nagoya University, Chikusa, Nagoya 464-8601, Japan. [3]Center for Gene Research, Nagoya University, Chikusa, Nagoya 464-8602, Japan. [4]Present address: Department of Bioscience and Biotechnology, Fukui Prefectural University, Eiheiji 910-1195, Japan. ✉e-mail: matsu@bio.nagoya-u.ac.jp

affinity of FLS2 for the flagellin monomer is far weaker than that for flg22 peptide[2,9], indicating that proteolytic release of the flg22 peptide from the flagellin protein is required for full induction of defense responses. However, in contrast to the well-characterized signaling cascade downstream of FLS2, little is known about the mechanisms of proteolytic release of the immunogenic peptide ligand flg22 from flagellin.

The importance of proteolytic digestion of flagellin in initiating the host immune response also is suggested by the finding that the secreted plant glycosidase β-galactosidase 1 (BGAL1) participates in flagellin deglycosylation and contributes to immunity in *Nicotiana benthamiana*[10]. BGAL1 removes *O*-glycans located on the outward-facing hydrophilic surface of the flagellin in flagellar filaments, facilitating exposure of the flg22 region to proteases that release the flg22 ligand. Consistent with this scenario, the addition of a protease inhibitor to apoplastic fluids containing BGAL1 blocks the release of immunogenic peptides from bacterial flagellin when the host is challenged with deglycanated flagellin protein, demonstrating an important role for proteases in immunogenic peptide release[10].

Bacteria enter the plant apoplast through natural openings such as stomata or wounds. The apoplast is the site of the initial interaction between plants and bacteria; thus, the proteases that cleave flg22 from flagellin likely are present in the apoplast. Proteolytic release of the flg22 peptide from the flagellin protein requires cleavage at both the N- and C-termini of the 22-amino-acid epitope. The *Arabidopsis* genome contains hundreds of protease-encoding genes, among which nearly 300 encode secreted proteases carrying signal peptides[11].

In the present study, we designed a fluorescence-quenching substrate to detect proteolytic activity at the C-terminus of the 22-amino-acid flg22 epitope in flagellin. Specifically, we established an efficient system for identification of the target proteases in *Arabidopsis* apoplastic fluid using native two-dimensional electrophoresis followed by an in-gel proteolytic assay and mass spectrometry-based proteomics. We identified two subtilases, SBT5.2 and SBT1.7, as specific proteases involved in C-terminal cleavage of flg22. The apoplastic fraction of *Arabidopsis* mutant plants deficient in these two proteases showed decreased C-terminal cleavage of flg22 from flagellin, resulting in a delay in apoplastic ROS production in leaf disks upon flagellin treatment.

## Results

### Proteolytic release of immunogenic peptides from bacterial flagellin by *Arabidopsis* apoplastic proteases

Proteolytic release of the immunogenic peptides from flagellin is thought to be required for full induction of host defense responses. To examine the temporal differences between the defense responses induced by the flagellin protein and the immunogenic flg22 peptide, we measured ROS production in wild-type (WT) *Arabidopsis* leaf disks treated with flagellin purified from *Pseudomonas syringae* pv. *tomato* DC3000 (*Pst* DC3000) or synthetic flg22 peptide from *Pst* DC3000. When WT leaf disks were exposed to protein or peptide at 100 nM, ROS production was delayed for ≈5 min following flagellin protein exposure compared to that induced by flg22 peptide (Fig. 1a, b). An *fls2* mutant (SAIL_691_C04) showed no ROS production upon exposure to either purified flagellin or synthetic flg22 (Supplementary Fig. 1a). We reasoned that this time lag represents the time taken for proteolytic release of the immunogenic peptide that acts as a ligand for FLS2.

The apoplast, as the primary interface between plants and bacteria, serves as the crucial space where flagellin fragmentation is most likely to occur. To investigate fragmentation patterns of flagellin in plant apoplasts, purified flagellin protein was incubated for 4 h with spent medium from *Arabidopsis* whole-plant submerged culture (submerged culture medium), into which secreted proteins (including proteases) that are present in the apoplast diffuse[12]. Of the secreted proteins detected in the actual leaf apoplast, 75% also were detected in

the submerged culture medium (Supplementary Data 1). Nano-liquid chromatography-tandem mass spectrometry (nano-LC-MS/MS) of the digests yielded ≈1600 peptide spectrum matches (PSMs) for peptide fragments derived from flagellin (Fig. 1c). We mapped all of the identified peptides against the amino acid sequence of flagellin and observed that multiple peptide fragments derived primarily from the N-terminal region of flagellin, including the flg22 epitope peptide, are generated by the *Arabidopsis* apoplastic proteases. In comparison, few peptide fragments were detected in mock-treated flagellin (Supplementary Fig. 1b).

Detailed inspection of the digestion patterns indicated that one of the major cleavage sites within flagellin is the C-terminus of the flg22 epitope, representing 20.5% of the total PSMs that were detected (Fig. 1d). In contrast, complex cleavage pattern was observed around the N-terminus of the flg22 epitope. We also detected cleavage activity in the central region of the flg22 epitope, which potentially causes loss of the immunogenic activity of flg22[2]; this observation suggested the existence of offensive and defensive interactions between plants and bacteria regarding cleavage of the flg22 epitope. Treatment of WT leaf disks with the synthetic flg22 peptide variant with a 6-residue C-terminal extension (flg22C6) resulted in delayed ROS production (Supplementary Fig. 1c, d); again, this observation suggested that the C-terminal cleavage of the flg22 epitope is important for activation of the defense response.

When flagellin was cleaved by the endoproteinase Asp-N, which cleaves peptide bonds N-terminal to Asp residues, peptide fragments were detected from across virtually the entire length of the protein, except for the $Ser^{143}$-$Ser^{201}$ domain to which carbohydrate chains are known to be attached (Supplementary Fig. 2a, b)[13]. These results indicated the presence of apoplastic proteases that predominantly cleave the flagellin N-terminal region, including those that release the flg22 epitope. Based on these results, we were interested in the molecular identity of the *Arabidopsis* endo-proteases that cleave flagellin to generate the C-terminal end of the flg22 epitope, and the extent to which these enzymes contribute to flagellin recognition by plants.

### Specific detection of cleavage at the C-terminus of flg22 using a fluorescence-quenching substrate

To identify the proteases that cleave the C-terminus of the flg22 domain within flagellin, we designed a fluorescence-quenching substrate that specifically detects proteolytic activity at the C-terminus of the flg22 domain; specifically, we used a 10-amino-acid peptide spanning the cleavage site (flg[47–56]). Fluorescence-quenching technology employs a paired fluorophore and quencher, which are introduced on the opposite ends of the peptide[14]. Upon hydrolysis of an internal peptide bond, the distance between the donor and the acceptor increases, the fluorophore is no longer quenched, and an enhanced fluorescence signal is generated. We synthesized flg[47–56] by replacing $Ser^{56}$ with a quencher, $N^{\varepsilon}$-2,4-dinitrophenyl-lysine (Lys(Dnp)), followed by introduction of a fluorophore, *N*-methylanthranic acid (Nma)[15], at the N-terminal amino group, to generate a fluorescence-quenching substrate, Nma-flg[47–56]-Dnp (Fig. 2a, b).

To test whether this substrate can detect the proteolytic activity at the C-terminus of flg22, we added 4 μM Nma-flg[47–56]-Dnp to the *Arabidopsis* submerged culture medium. The fluorescence intensity in the reaction increased over time during the 1–7 h after substrate addition, and the intensity maintained maximum levels for up to 20 h; in contrast, no increase was observed when we added the substrate to heat-inactivated submerged culture medium in which the proteases presumably had been denatured (Fig. 2c). Nano-LC-MS analysis of the digests confirmed that the major proteolytic product is a 5-amino-acid peptide cleaved at the C-terminus of the flg22 domain (Fig. 2d, e, Supplementary Fig. 3a, b). These results indicated that the fluorescence-quenching substrate Nma-flg[47-56]-Dnp can detect the *Arabidopsis* apoplastic endo-protease(s) that cleave the C-terminus of the flg22 domain within flagellin.

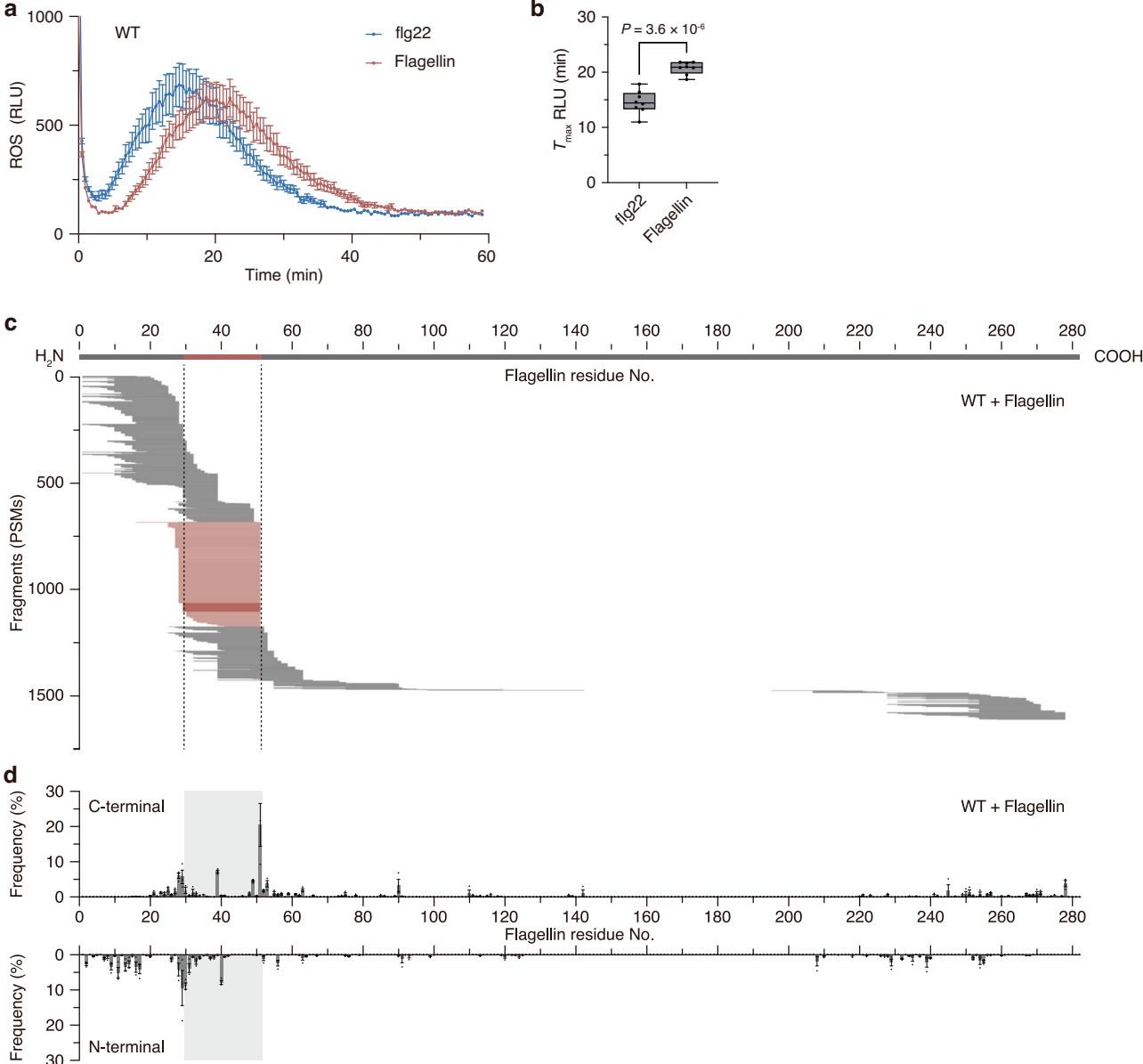

**Fig. 1 | Proteolytic release of immunogenic flg22 peptides from flagellin protein. a** ROS burst in *Arabidopsis* leaves upon elicitor exposure. Leaf disks from WT plants were incubated with 100 nM flg22 peptide or flagellin protein. ROS production was measured every 30 s over 60 min. Values represent the mean ± SE ($n = 8$ biologically independent samples). **b** Comparison of time at which the maximum response ($T_{max}$) was observed ($n = 8$ biologically independent samples). Boxes represent 25th to 75th percentile range, the line within the box marks the median and the whiskers represent the minimum and maximum values. *P* value was calculated by two-tailed non-paired Student's *t*-test. **c** Representative fragmentation pattern of flagellin upon incubation in *Arabidopsis* submerged culture medium containing apoplastic proteases. Flagellin protein was incubated for 4 h in WT

*Arabidopsis* submerged culture medium, and the resulting peptides were analyzed by nano-LC-MS/MS. Each PSM was mapped against the amino acid sequence of flagellin. The thick line at the top represents flagellin protein; the red domain indicates the flg22 epitope. PSMs that exactly match flg22 peptide are shown as thin dark-red lines. PSMs that match fragments cleaved at the C-terminus of the flg22 domain are shown as thin light-red lines. **d** Frequency at which each amino acid residue is the C-terminal end (upper panel) or N-terminal end (lower panel) of the peptides following digestion by WT apoplastic proteases. The gray box indicates the flg22 epitope. Values represent the mean ± SE ($n = 3$ biologically independent samples). Source data are provided as a Source Data file.

## Identification of endo-proteases involved in C-terminal cleavage of flg22

The fluorescence-quenching substrate Nma-flg[47-56]-Dnp enabled high-throughput fluorogenic detection of proteolytic activity at the C-terminus of flg22. We employed this system for in-gel detection of endo-proteases on polyacrylamide gels. *Arabidopsis* submerged culture medium was concentrated and subjected to native two-dimensional electrophoresis. Proteins were separated in the first dimension by native isoelectric focusing (IEF) and in the second dimension by blue native polyacrylamide gel electrophoresis (BN-

PAGE), followed by gel fractionation using a SAINOME-plate, which dices the gel into approximately 4.5-mm square pieces and drops these gel fragment into the individual wells of a 384-well plate[16]. We then performed an in-gel proteolytic assay in the 384-well plate using 40 μM Nma-flg[47-56]-Dnp as a substrate and detected any active spots with increased fluorescence (Fig. 3a).

To identify the proteins in this spot, we performed mass spectrometry-based proteomics of gel pieces at and around the spot at G9, which showed the highest activity; those at spot H14 were analyzed as a negative control. The resulting MS/MS data were used to search a

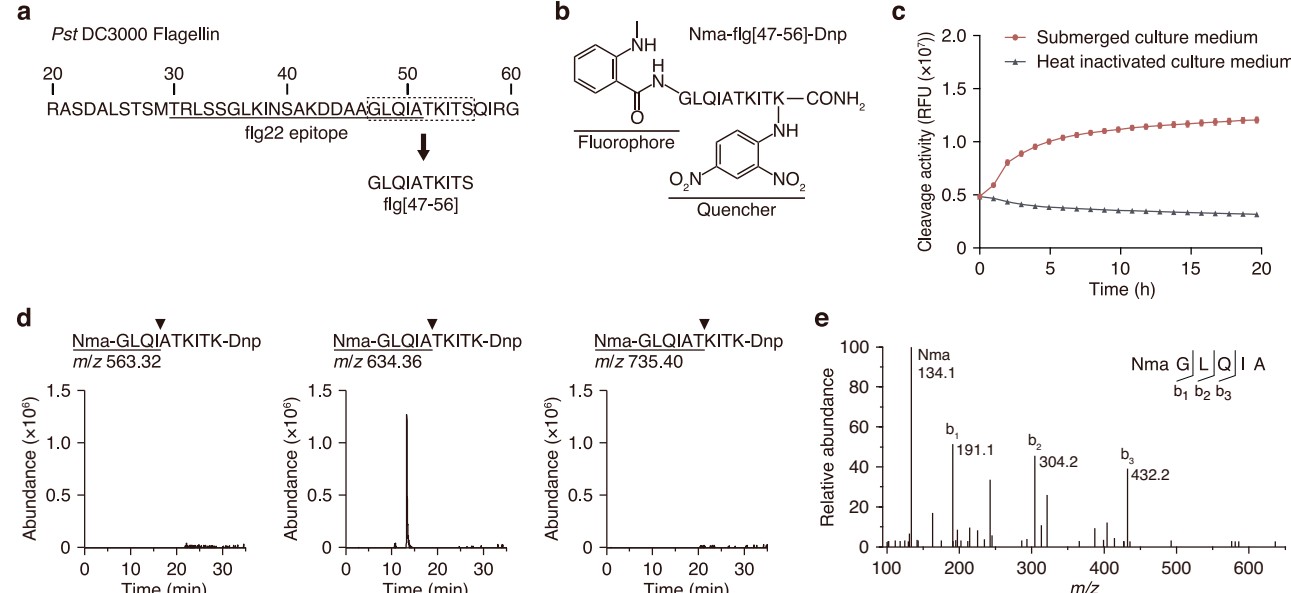

**Fig. 2 | Detection of cleavage activity at the C-terminus of the flg22 epitope within flagellin by fluorescence-quenching substrate. a** The amino acid sequence near the N-terminus of *Pst* DC3000 flagellin. The underlining indicates the flg22 epitope domain; the dashed box indicates the flg[47–56] sequence used to construct the substrate. **b** Structure of the fluorescence-quenching substrate Nma-flg[47-56]-Dnp. **c** Fluorescence intensity of Nma-flg[47-56]-Dnp during incubation with *Arabidopsis* submerged culture medium (red circles) or heat-inactivated submerged culture medium (gray triangles). Fluorescent emission at 435 nm was measured every 1 h over 20 h. Values represent the mean ± SE ($n = 3$ biologically independent samples). **d** In vitro proteolytic assay of Nma-flg[47–56]-Dnp with *Arabidopsis* submerged culture medium. Representative selected ion chromatograms (SICs) are shown for Nma-GLQI, Nma-GLQIA, and Nma-GLQIAT. **e** MS/MS spectrum of the precursor ion at *m/z* 634.36 corresponding to Nma-GLQIA. Source data are provided as a Source Data file.

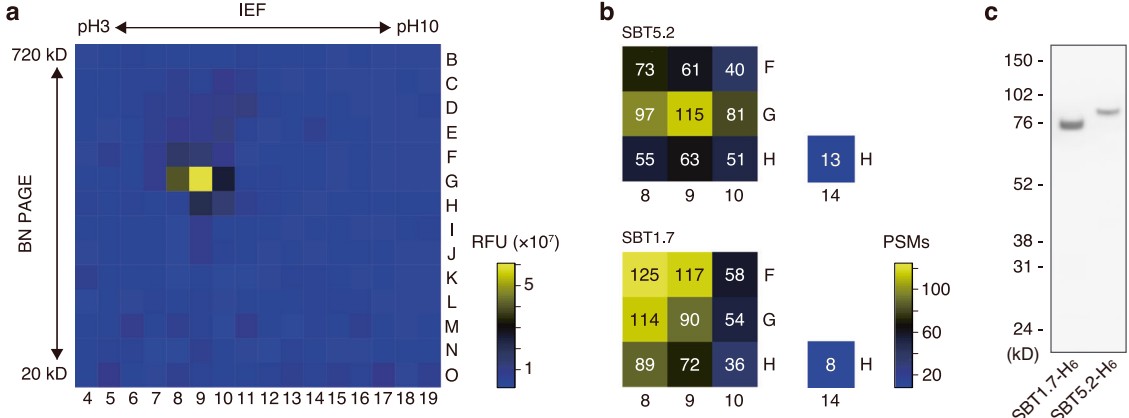

**Fig. 3 | Identification of SBT5.2 and SBT1.7 as endo-proteases involved in C-terminal cleavage of the flg22 domain. a** In-gel proteolytic assay of the diced two-dimensional gels using Nma-flg[47–56]-Dnp as a fluorescence-quenching substrate. The fluorescence intensity was measured 6 h after the addition of the substrate. The numbers and letters represent the columns and rows (respectively) of the microplate used in the assay. The fluorescence intensity is color-coded as indicated in the scale on the right. **b** Total count of PSMs of SBT5.2 and SBT1.7 identified in the fluorescence-positive gel fragments and in the fluorescence-negative gel fragment used as a control. The PSM count is color-coded as indicated in the scale on the right. **c** *Agrobacterium*-mediated transient expression of the His-tagged subtilases, SBT5.2-H$_6$ and SBT1.7-H$_6$, in *N. benthamiana*. Apoplastic fluids were extracted from agroinfiltrated leaves and subjected to SDS-PAGE followed by western blotting using HRP-conjugated anti-His tag antibody. The experiment was independently repeated three times with similar results. Source data are provided as a Source Data file.

database of *Arabidopsis* proteins, and proteases with an N-terminal signal peptide were extracted from among the identified proteins. This procedure identified two subtilases, SBT5.2 (At1g20160) and SBT1.7 (At5g67360; also known as ARA12), both of which were detected specifically in active gel fractions with high PSM scores (Fig. 3b, Supplementary Data 2).

### SBT5.2 and SBT1.7 cleave the C-terminus of flg22 from flagellin

To test whether SBT5.2 and SBT1.7 cleave at the C-terminus of the flg22 domain, we transiently expressed C-terminally His-tagged SBT5.2 or

SBT1.7 (SBT5.2-H$_6$ or SBT1.7-H$_6$, respectively) by agroinfiltration of *N. benthamiana*; we then isolated apoplastic fluid from the agroinfiltrated plants. The expression of SBT5.2-H$_6$ and SBT1.7-H$_6$ was confirmed by western blot analysis using anti-His-tag antibody (Fig. 3c). We captured SBT5.2-H$_6$ or SBT1.7-H$_6$ proteins on Ni-Sepharose beads for use in solid-phase protease assays. Incubation of flagellin protein with the beads for 1 h followed by nano-LC-MS/MS analysis confirmed that both SBT5.2-H$_6$ and SBT1.7-H$_6$ cleave precisely at the C-terminus of the flg22 domain (Ala[51]-Thr[52]) (Fig. 4a–e). Alanine-scanning experiments demonstrated that a residue at the P2 position (Ile[50]) is critical

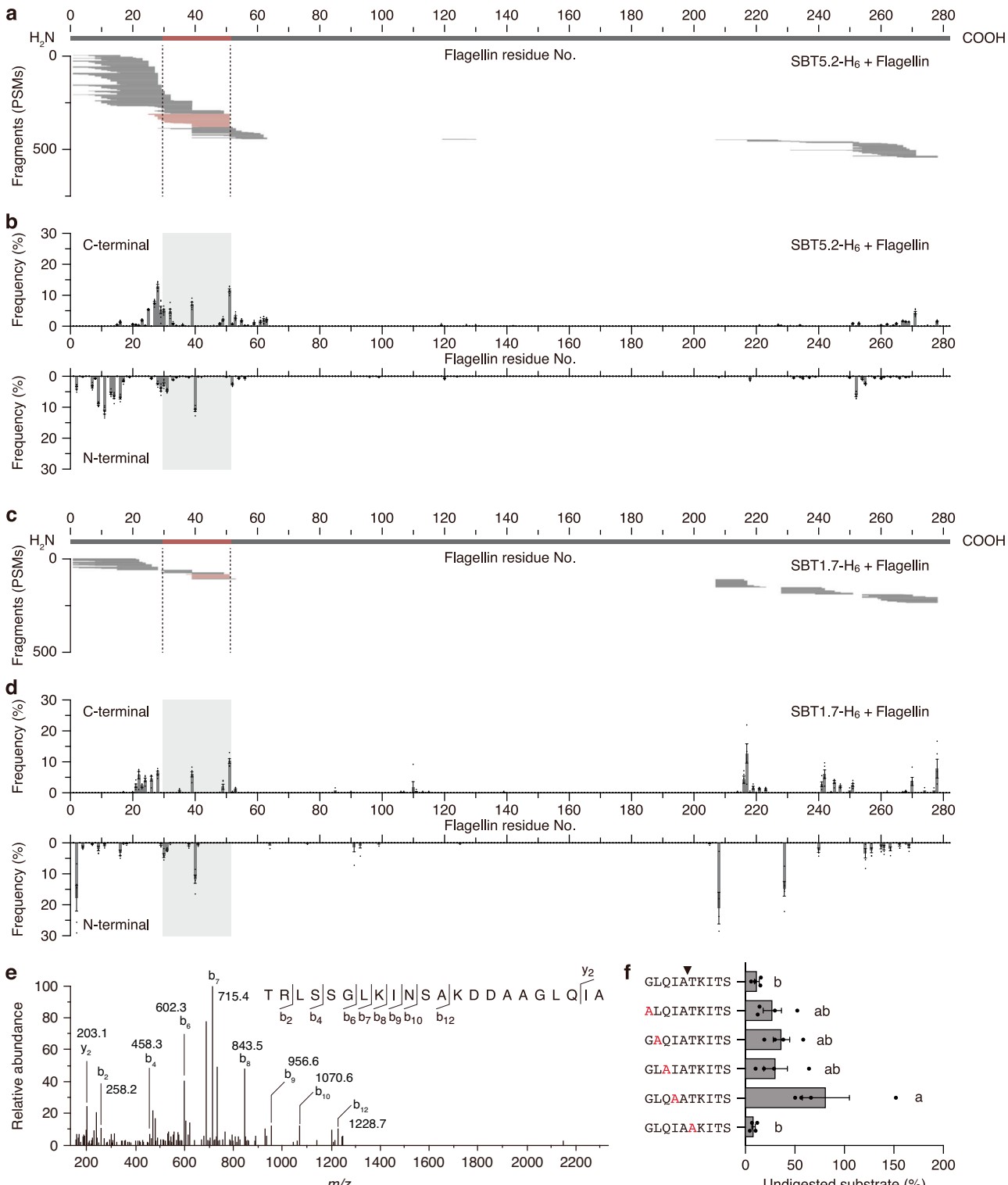

**Fig. 4 | Cleavage patterns of flagellin digested with SBT5.2-H$_6$ and SBT1.7-H$_6$.**
**a** Representative fragmentation pattern of flagellin digested with His-tagged SBT5.2 (SBT5.2-H$_6$). Flagellin protein was incubated for 1 h with bead-captured SBT5.2-H$_6$, and the resulting peptide fragments were analyzed by nano-LC-MS/MS. Each PSM was mapped against the amino acid sequence of flagellin. The thick line at the top represents flagellin protein; the red domain indicates the flg22 epitope. PSMs that matched fragments cleaved at the C-terminus of flg22 domain are shown as red lines. **b** Frequency at which each amino acid residue is the C-terminal end or N-terminal end of the peptides following digestion with SBT5.2-H$_6$. The gray box indicates the flg22 epitope. Values represent the mean ± SE ($n$ = 6 biologically independent samples). **c** Representative fragmentation pattern of flagellin by

SBT1.7-H$_6$. **d** Frequency at which each amino acid residue is the C-terminal end or N-terminal end of the peptides following digestion by SBT1.7-H$_6$. Values represent the mean ± SE ($n$ = 5 biologically independent samples). **e** Representative MS/MS spectrum of the flg22 epitope peptide cleaved from flagellin by SBT5.2-H$_6$. **f** Effect of alanine-scanning substitution of the C-terminal cleavage site of flg22 on cleavage activity by SBT5.2. Each synthetic peptide was incubated for 1 h with bead-captured SBT5.2-H$_6$, and the remaining undigested peptides were quantified by nano-LC-MS. Black arrowhead indicates cleavage site. Different letters indicate a significant difference with others (mean ± SE, $P$ < 0.05, one-way ANOVA followed by Tukey's test, $n$ = 4 biologically independent samples). Source data are provided as a Source Data file.

for efficient cleavage by SBT5.2-$H_6$ at this site (Fig. 4f). When flagellin was digested by SAP2 (AT1G03220), a secreted protease detected nonspecifically in gel fractions (Supplementary Data 2), the Ala[51]-Thr[52] site was not cleaved, indicating the substrate specificity of SBT5.2 and SBT1.7 (Supplementary Fig. 4a, b).

In addition to the effects of the subtilases on Ala[51]-Thr[52] cleavage, SBT5.2-$H_6$ and SBT1.7-$H_6$ provided proteolytic cleavage of flagellin at Asn[39]-Ser[40], a site located in the central region of the flg22 domain (Fig. 4a–d). At this cleavage site, the residue at the P2 position is also Ile. Since the flg22 epitope loses its immunogenic activity when cleaved at Asn[39]-Ser[40] [2], the balance of the cleavage efficiency between Asn[39]-Ser[40] and Ala[51]-Thr[52] is critical for liberation of active immunogenic peptides.

We also detected minor but multiple cleavage sites within the N-terminal region of flagellin (Fig. 4a–d). We confirmed that flagellin does not fragment when incubated with Ni-Sepharose beads exposed to control apoplastic fluid collected from *N. benthamiana* plants expressing the empty vector (Supplementary Fig. 4c). We further confirmed that immobilized *Arabidopsis* SBT5.2-$H_6$ and SBT1.7-$H_6$ contain few or no detectable contamination by *N. benthamiana* proteases, thus excluding the possibility that multiple cleavage of the N-terminal region of flagellin reflects co-purified *N. benthamiana* proteases (Supplementary Data 3). The cleavage sequence preferences of SBT5.2 and SBT1.7 are similar, but not identical, to each other. Unlike SBT5.2-$H_6$, SBT1.7-$H_6$ was able to efficiently cleave the C-terminal region of flagellin at sites such as Asn[207]-Ser[208], Gln[217]-Asn[218] and Gln[228]-Asn[229] (Fig. 4c, d). These results indicated that SBT5.2 and SBT1.7 have preferential cleavage target sequences, but selection of the cleavage site is not always stringent.

The flg22 domain is highly conserved in bacterial flagellin, but there is some sequence variation around the C-terminus of the flg22 domain depending on the bacterial species[2]. Therefore, we further investigated whether SBT5.2 cleaves the C-terminus of the flg22 domain of flagellins derived from bacterial species other than *Pst* DC3000. Flg22 peptides derived from the flagellin proteins of *Pseudomonas aeruginosa* and *Xanthomonas axonopodis* pv. *citri* both exhibit PTI-inducing activity[2,17]. Based on the flagellin sequences of these two bacterial species, two separate 10-amino-acid-long peptides, flg[47-56][Pa] and flg[47-56][Xac] (respectively) were synthesized; each peptide then was incubated for 1 h with SBT5.2-$H_6$. Nano-LC-MS analysis confirmed that SBT5.2-$H_6$ cleaves both flg[47-56][Pa] and flg[47-56][Xac] at the C-terminus of the flg22 domain, exhibiting cleavage efficiencies comparable to that observed with flg[47-56] (Supplementary Fig. 5a–f). These data suggested that SBT5.2 functions against substrates from across bacterial species.

### Decrease in C-terminal cleavage of flg22 from flagellin in *sbt5.2 sbt1.7* double mutant

To determine the function of SBT5.2 and SBT1.7 *in planta*, an *sbt5.2 sbt1.7* double mutant was generated by crossing the T-DNA insertion mutant lines for *SBT5.2* (SALK_132812C, *sbt5.2-2*) and *SBT1.7* (GABI_544E06, *sbt1.7-2*) (Fig. 5a). RT-PCR confirmed that full-length *SBT5.2* and *SBT1.7* transcripts were absent in the *sbt5.2 sbt1.7* double mutant (Fig. 5b).

To investigate changes in flagellin fragmentation patterns in the apoplast of *sbt5.2 sbt1.7* plants, purified flagellin protein was incubated for 4 h with submerged culture medium obtained from WT, *sbt5.2* single, *sbt1.7* single or *sbt5.2 sbt1.7* double-mutant plants. Nano-LC−MS/MS of the digests exposed to the WT submerged culture medium revealed that as much as 15.8% PSMs were derived from peptide fragments cleaved at the C-terminus of the flg22 domain (Fig. 6a, b). In contrast, when incubated with submerged culture medium of the *sbt5.2 sbt1.7* double mutant, only 0.6% of the total PSMs corresponded to fragments cleaved at the C-terminus of flg22 domain, confirming that proteolytic activity at the C-terminus of the flg22 domain is markedly decreased in the *sbt5.2 sbt1.7* mutant (Fig. 6c, d).

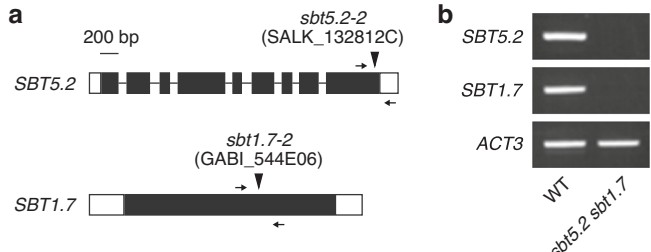

**Fig. 5 | Characterization of the T-DNA insertion mutants for *SBT5.2* and *SBT1.7*. a**- The T-DNA insertion sites of the *sbt5.2-2* and *sbt1.7-2* mutants. Exons are indicated as filled gray rectangles, introns are indicated as solid lines, and untranslated regions are indicated as white rectangles. Arrows indicate the sites of primers used for RT-PCR analysis. **b** Absence of *SBT5.2* and *SBT1.7* transcripts in the *sbt5.2 sbt1.7* double mutant, as verified by RT-PCR. *ACT3* was used as an internal control. Source data are provided as a Source Data file.

We also observed decreased proteolytic activity at the C-terminus of the flg22 domain in submerged culture medium from the *sbt5.2* single mutant, compared to that of WT (Supplementary Fig. 6a–c). In contrast, the flagellin digestion pattern observed in submerged culture medium from the *sbt1.7* single mutant was similar to that obtained from the WT (Supplementary Fig. 6a, b, d). Submerged culture medium from a complementation line, in which the *SBT5.2* gene was introduced into the *sbt5.2 sbt1.7* double mutant, showed proteolytic cleavage patterns comparable to those observed with the submerged culture medium from *sbt1.7* single-mutant or WT plants (Supplementary Fig. 6a, b, e). These results indicated that SBT5.2 is the major apoplastic protease responsible for the cleavage of flagellin at the C-terminus of the flg22 domain *in planta*, and that SBT1.7 also functions redundantly in the generation of this immunogenic peptide.

### Loss of SBT5.2 and SBT1.7 delays ROS production in leaf disks following flagellin treatment

To investigate the biological role of SBT5.2- and SBT1.7-mediated cleavage of flagellin during PTI, we measured ROS production in leaf disks of WT and *sbt5.2 sbt1.7* double-mutant plants treated with 100 nM flagellin protein. We found that ROS production was delayed for ≈2 min in *sbt5.2 sbt1.7* leaf disks, compared to WT leaf disks, following flagellin treatment, although the maximum signal intensity was comparable between the WT and *sbt5.2 sbt1.7* double-mutant plants, presumably due to residual protease activity (Fig. 7a–d). This time lag between WT and *sbt5.2 sbt1.7* in response to flagellin treatment is reminiscent of the time lag observed in leaf disks of WT plants in response to the synthetic flg22 peptide and flagellin (Fig. 1a, b). We confirmed that there was no time lag in ROS production between WT and *sbt5.2 sbt1.7* when each was treated with 100 nM synthetic flg22 peptide, demonstrating that there is no difference in sensitivity to cleaved flg22 between WT and *sbt5.2 sbt1.7* plants (Fig. 7b, c).

We interpreted this phenotype as being a consequence of the reduced proteolytic activity at the C-terminus of the flg22 domain in *sbt5.2 sbt1.7*. Loss of *SBT5.2* and *SBT1.7* would increase the time required for proteolytic release of the immunogenic peptides, resulting in delayed ROS production.

## Discussion

In the present study, we analyzed the fragmentation patterns of flagellin in *Arabidopsis thaliana* apoplastic fluid using nano-LC−MS/MS to understand proteolytic release of the immunogenic peptide from flagellin protein. Detailed inspection of the flagellin digestion patterns revealed that one of the major endo-cleavage sites within flagellin protein is the C-terminus of the flg22 domain. Furthermore, by using a proteomic approach in combination with a fluorescence-quenching peptide substrate, we identified two subtilases, SBT5.2 and SBT1.7, as

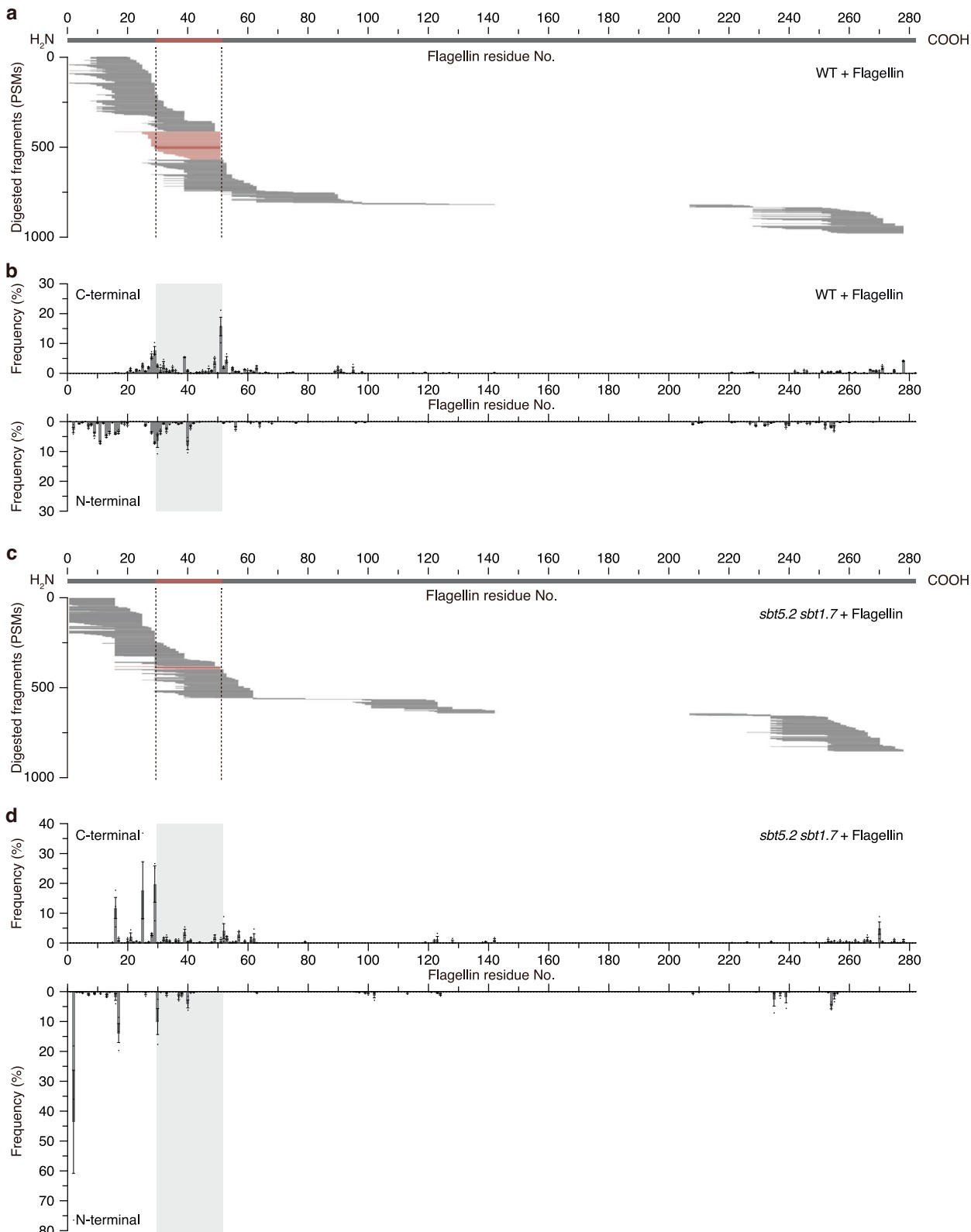

**Fig. 6 | The *sbt5.2 sbt1.7* double mutant exhibits decreased C-terminal cleavage of the flg22 domain from flagellin. a** Representative fragmentation pattern of flagellin following incubation for 4 h in submerged culture medium from WT plants. **b** Frequency at which each amino acid residue is the C-terminal or N-terminal end of the peptides following digestion of flagellin by submerged culture medium from WT plants. Values represent the mean ± SE (*n* = 3 biologically independent samples). **c** Representative fragmentation pattern of flagellin following incubation for 4 h in submerged culture medium from *sbt5.2 sbt1.7* plants. **d** Frequency at which each amino acid residue is the C-terminal or N-terminal end of the peptides following digestion of flagellin by submerged culture medium from *sbt5.2 sbt1.7* plants. Values represent the mean ± SE (*n* = 3 biologically independent samples). Source data are provided as a Source Data file.

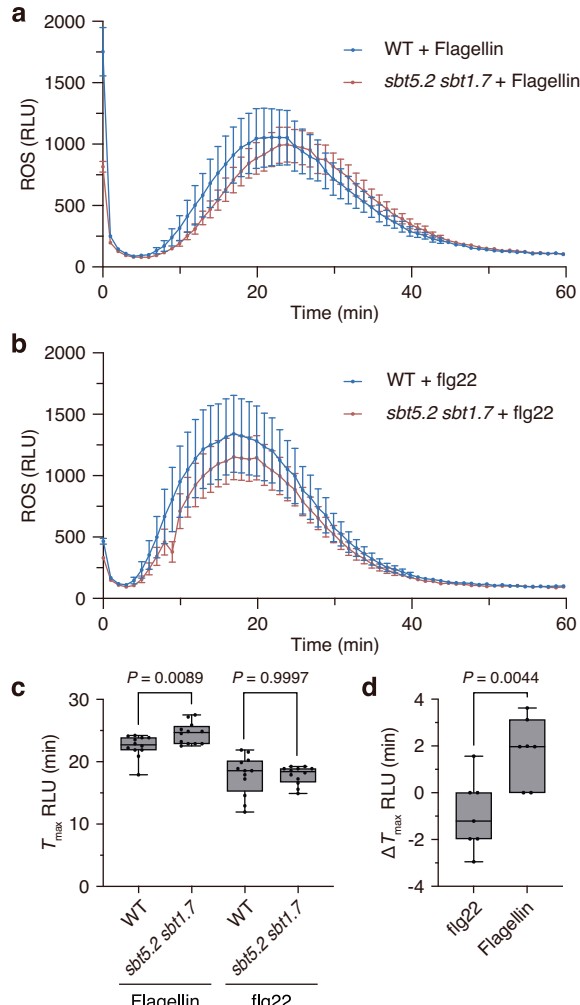

**Fig. 7 | Loss of SBT5.2 and SBT1.7 delays ROS production in leaf disks following flagellin exposure. a** ROS burst in WT and *sbt5.2 sbt1.7* leaf disks following exposure to 100 nM flagellin. ROS production was measured every minute over 60 min. Values represent the mean ± SE ($n = 12$ biologically independent samples). **b** ROS burst in WT and *sbt5.2 sbt1.7* leaf disks following exposure to 100 nM flg22. Values represent the mean ± SE ($n = 12$ biologically independent samples). **c** Comparison of time at which the maximum response ($T_{max}$) observed in (**a**) and (**b**) ($n = 12$ biologically independent samples). **d** Delay in time to maximum RLU ($\Delta T_{max}$ RLU) of the *sbt5.2 sbt1.7* double mutant compared to that of WT in seven independent experiments ($n = 7$ independent experiments). For all box plots, boxes represent 25th to 75th percentile range, the line within the box marks the median and the whiskers represent the minimum and maximum values. *P* values were calculated by two-tailed non-paired Student's *t* test. Source data are provided as a Source Data file.

proteases involved in flagellin cleavage at the C-terminus of the flg22 domain. It has been reported in *Nicotiana benthamiana* that the secreted glycosidase BGAL1 is involved in the deglycosylation of flagellin, facilitating the exposure of the flg22 region to proteases that release the flg22 ligand[10]. Because At2g28470, the *Arabidopsis* homolog of tobacco BGAL1, is also detected in submerged culture medium (Supplementary Data 1), it may facilitate flg22 cleavage. However, since purified SBT5.2-H$_6$ alone can cleave flagellin, it appears that flagellin is cleaved to some extent even in the absence of BGAL1.

The proteolytic activity of SBT5.2 and SBT1.7 in cleaving flagellin at the C-terminus of flg22 was confirmed not only for the amino acid sequence of flagellin from *P. syringae* pv. *tomato* DC3000, but also for those from *P. aeruginosa* and *Xanthomonas axonopodis* pv. *citri*, two additional species that generate flg22 peptides that induce PTI[2,17]. These data suggest that SBT5.2 and SBT1.7 possess activity against

substrates from across bacterial species. The broad substrate specificity of the flg22 cleavage proteases against substrates from across bacterial species may ensure that FLS2 recognizes a wide range of pathogenic bacteria.

In *sbt5.2 sbt1.7* double-mutant plants, the apoplastic C-terminal cleavage of flg22 from flagellin was decreased, resulting in delayed ROS production in leaf disks upon flagellin treatment. A recent study has highlighted the significance of apoplastic proteases in the release of immunogenic peptides[10]. The addition of protease inhibitors to the apoplastic fluid inhibits the release of immunogenic peptides from bacterial flagellin, leading to the suppression of ROS production in leaf disks[10]. Plants lacking SBT5.2 and SBT1.7, however, did not achieve complete suppression of ROS production activity. We attribute this incomplete attenuation to the presumed involvement of additional proteases, other than SBT5.2 and SBT1.7, in the release of immunogenic peptides from flagellin. Indeed, the flagellin cleavage activity at the C-terminus of the flg22 domain was decreased, rather than completely abrogated, in the *sbt5.2 sbt1.7* double-mutant plants. These results suggest the existence of other proteases that cleave flagellin at the C-terminus of flg22. Furthermore, in the apoplast of *Arabidopsis*, particularly in that of *sbt5.2 sbt1.7* double-mutant plants, peptide fragments with C-terminal extensions from the C-terminus of the flg22 domain also were detected. The elucidation of the potential ligand activity of these C-terminally extended peptides and the proteases involved in their cleavage remains a challenge. Additionally, the cleavage of the N-terminus of the flg22 domain is necessary for the release of the immunogenic peptides. The fragmentation pattern of flagellin in submerged culture medium obtained from *sbt5.2 sbt1.7* plants suggests the presence of endo-proteases that cleave precisely at the N-terminus of the flg22 domain.

Previous work has shown that SBT5.2 and SBT1.7 contribute to various physiological processes by cleaving precursors of endogenous peptide hormones as substrates in the apoplast. SBT5.2 has been reported to be involved in the precursor processing and maturation of the IDA (INFLORESCENCE DEFICIENT IN ABSCISSION) peptide, which controls floral organ abscission[18]; the SBT5.2 enzyme also was identified as a protease (referred to as CRSP (CO$_2$ RESPONSE SECRETED PROTEASE)) that cleaves and activates the pro-peptide EPF2 (EPIDERMAL PATTERNING FACTOR 2), which in turn represses stomatal density[19]. Separately, SBT1.7 has been reported to function in the precursor processing of the CLE40 (CLAVATA3/EMBRYO SURROUNDING REGION-RELATED 40) peptide, which controls stem cell maintenance in the root[20]. In addition to known functions of these enzymes in cleaving endogenous proteins, the present study demonstrates that these proteases cleave flagellin, an exogenous protein from pathogenic bacteria. This finding demonstrates the dual role of these subtilases in regulating both internal and external signals.

*SBT5.2* has two splice variants, *SBT5.2(a)* and *SBT5.2(b)*. SBT5.2(a) is secreted to the apoplast and exhibits protease activity; in contrast, SBT5.2(b) lacks a signal peptide, preventing its secretion, and this protein does not exhibit protease activity[21]. SBT5.2(b) localizes to the endosomes and interacts with the defense-related transcription factor MYB30 (MYELOBLASTOSIS-LIKE PROTEIN 30), resulting in the retention of MYB30 at the endosomes and exclusion from the nucleus. As a result, SBT5.2(b) negatively regulates the defense-related cell death induced by *Pst AvrRpm1*[21]. Our study revealed that SBT5.2(a) functions upstream of flagellin perception and positively regulates PTI. Recent studies have demonstrated that PTI is required for effector-triggered hypersensitive cell death, and PTI enhances cell death induced by effector-triggered immunity (ETI)[22,23]. These findings suggest that SBT5.2(a) and SBT5.2(b) play opposing roles in defense responses associated with cell death. Presumably, defense responses need to be tightly controlled, given that such responses often are accompanied by growth inhibition. To maintain this balance, *Arabidopsis* may utilize two splice variants of the *SBT5.2* gene with opposing functions.

Alternatively, another scenario can be used to explain the existence of these two opposing splice variants. It has been reported that during bacterial treatment, host expression of *SBT5.2(a)* is down-regulated, while the expression of *SBT5.2(b)* is induced[21]. If this expression change is not a control mechanism employed by the plant to balance immunity but rather controlled by the bacteria, SBT5.2 may be acting as a suppressor of immunity through splicing regulation, which would benefit the bacteria. Therefore, the bacteria may not try to evade the cleavage of flagellin by SBT5.2(a) and the resulting recognition. This hypothesis may explain why splice variants of this protease with opposing functions have persisted evolutionarily.

It is also evolutionarily intriguing that SBT5.2 and SBT1.7 are not only involved in flg22 peptide liberation by cleavage of Ala$^{51}$-Thr$^{52}$, which is beneficial to the plant, but also in cleavage of Asn$^{39}$-Ser$^{40}$, which leads to inactivation of the flg22 peptide and benefits the bacterial side. An Ile residue required for SBT5.2 recognition is also found at the P2 position of the Asn$^{39}$-Ser$^{40}$ cleavage site, and this Ile-Asn-Ser sequence is conserved in the flg22 domain of many flagellin proteins[2]. Comprehensive amino acid substitution experiments in the flg22 domain of flagellin have revealed that 79.8% of the mutants lose motility, while 74.5% of the mutants still maintain binding activity to FLS2[24]. This means that it would be difficult for bacteria to evolve the flg22 domain to escape recognition by FLS2 without losing motility. Therefore, rather than altering the overall conformation of the flg22 peptide to reduce its binding ability to FLS2, bacteria may have evolved to allow the plant enzyme to cleave one of the regions required for flg22 activity, thereby escaping FLS2 recognition and improving their own fitness. There seems to be an evolutionary "hide-and-seek" between plants and bacteria over which site of flg22, Asn$^{39}$-Ser$^{40}$ or Ala$^{51}$-Thr$^{52}$, is preferentially cleaved.

Even when a protein cleavage site has been identified or deduced, it is often the case that the corresponding protease has not yet been found. For example, although the proteolytic release of another flagellin epitope (flgII-28, which is recognized by FLS3[25–27]), also is expected, the proteases responsible for this cleavage have not yet been identified. It is also known that a number of peptide hormones are produced from larger inactive precursor proteins by proteolytic processing. Recent reports have identified some of the proteases involved in these processes[28,29], but there are still many examples where the corresponding enzymes have not yet been identified. The method employed in the present study, using fluorescence-quenching substrates in combination with native two-dimensional electrophoresis and proteomics, may be helpful for the identification and characterization of such proteases.

## Methods

### Plant materials and growth conditions

*Arabidopsis thaliana* ecotype Columbia (Col) was used as the wild type (WT). The *sbt5.2-2* (SALK_132812C), *sbt1.7-2* (GABI_544E06), and *fls2* (SAIL_691_C04) mutants, which were described previously[4,21,30] and obtained from the Arabidopsis Biological Resource Center. The double mutant with homozygous T-DNA insertions in both *SBT5.2* and *SBT1.7* was generated by crossing the single mutants. The homozygosity of the mutation was confirmed by RT-PCR using total RNA isolated from *Arabidopsis* leaves using specific primers (Supplementary Data 4). Total RNA was extracted using the RNeasy Mini Kit (QIAGEN) according to the manufacturer's protocol. Plants were grown from surface-sterilized *Arabidopsis* seeds that were vernalized for 2 d at 4 °C, sown on B5 medium containing 1.0% sucrose solidified with 0.7% agar, and incubated under continuous light at 22 °C. For the ROS production assay, plants were transplanted to soil and grown under a 8-h/16-h light/dark cycle at 22 °C. For the whole-plant submerged culture, forty 6-day-old seedlings were transplanted into 100 mL of B5 liquid medium containing 1.0% sucrose and incubated without shaking under continuous light at 22 °C for 7–11 days. For agroinfiltration,

*N. benthamiana* plants were grown on soil under a 16-h/8-h light/dark cycle at 23 °C for 4–5 weeks.

### Peptide synthesis

Epitope peptide flg22 from *Pst* DC3000 (TRLSSGLKINSAKDDAAG LQIA) and its variant flg22C6 (TRLSSGLKINSAKDDAAGLQIATKITSQ) were synthesized by fluorenylmethoxycarbonyl protecting group (Fmoc) solid-phase chemistry on Wang-PEG resin using an automated peptide synthesizer (Biotage Initiator+ Alstra). Using the same method, the peptide substrate flg[47–56] (GLQIATKITS), its Ala-substituted peptides, flg[47-56]$^{Pa}$ (GLQIANRLTS) and flg[47−56]$^{Xac}$ (GLAISERFTT) were synthesized. All peptides were purified by reverse-phase HPLC. Nma-flg[47–56]-Dnp was synthesized by the Peptide Institute, Inc. (Osaka, Japan).

### Preparation of flagellin

*Pst* DC3000 was grown on King's B agar plate containing 100 μg/mL rifampicin at 28 °C, then inoculated into King's B liquid medium containing rifampicin and incubated overnight at 28 °C with shaking at 150 rpm. A portion of this culture was inoculated into ≈40 volumes of fresh King's B liquid medium containing rifampicin and incubated overnight at 28 °C with shaking at 150 rpm. The bacterial pellet was collected by centrifugation at 3000 × *g* for 10 min and washed three times with 20 mM Tris-HCl (pH 8.0). To shear the flagella, the cell suspension was intensively blended for 2 min at 8000 rpm using a Polytron PT 10/35 homogenizer, and then centrifuged at 5000 × *g* for 10 min to pellet the cells[31]. The resulting supernatant was centrifuged at 100,000 × *g* for 1 h to pellet the flagella, and the resulting pellet then was suspended in 0.1 M glycine-HCl buffer (pH 2.0) for dissociation[32]. The flagellin protein was separated by a second round of centrifugation at 100,000 × *g* for 1 h, and the pH of the resulting supernatant (flagellin solution) was adjusted to 7.0 with NaOH.

### Measurement of ROS production

ROS production was measured as described previously[33]. Briefly, using a 4-mm-diameter cork borer, leaf disks were excised from soil-grown *Arabidopsis* leaves and individual punches were floated, adaxial side up, in 200 μL water per well in a 96-well white plate; the plate then was incubated overnight at room temperature. On the following day, the water was replaced with 100 μL of an aqueous suspension containing 100 nM flg22, flg22C6 or flagellin, 34 μg/mL luminol (Sigma-Aldrich), and 20 μg/mL horseradish peroxidase (HRP; Sigma-Aldrich). The relative luminescence unit (RLU) of each well was measured using a VICTOR Nivo Multimode Plate Reader (PerkinElmer) for 60 min. For the analysis of ROS production delay, $T_{max}$ RLU was defined as the time to reach the maximum RLU, and $\Delta T_{max}$ RLU was obtained by subtracting $T_{max}$ RLU of WT from that of *sbt5.2 sbt1.7*. Statistical analyses were performed using Prism software (version 9; GraphPad).

### Flagellin fragmentation in *Arabidopsis* submerged culture medium

Submerged culture medium was collected by decantation. Flagellin (final concentration, 100 nM) was added to 1 mL of submerged culture medium, and the mixture was incubated without shaking under continuous light at 22 °C for 4 h. Digested peptides were extracted from the incubated mixture using *o*-chlorophenol as described previously[12], with some modifications. Briefly, N-ethylmorpholine (NEM) (final concentration, 1.0%) and 500 μL of *o*-chlorophenol saturated with 1% NEM were added to each sample. After shaking for 1 min at room temperature, each sample was centrifuged at 9390 × *g* for 20 min. The aqueous phase was removed and the phenolic phase was recentrifuged at 9390 × *g* for 10 min. The phenolic phase then was collected, and peptides were precipitated with 10 volumes of acetone and 0.1% Ficoll at −30 °C overnight. After centrifugation at 9390 × *g* for 10 min, the resulting pellet was rinsed twice with acetone and then dried *in vacuo*.

The precipitate then was dissolved in 10 μL of water. This solution was diluted 20-fold with 2% acetonitrile (containing 0.1% TFA) and then desalted using a GL-Tip SDB (GL Science) according to the manufacturer's instructions. After vacuum concentration, the samples were dissolved in 20 μL of 2% acetonitrile (containing 0.1% TFA) and entire 20 μL was analyzed by nano-LC-MS/MS.

## Flagellin fragmentation by Asp-N

For Asp-N digestion, 3 μg of flagellin was reconstituted in 17 μL of 2 M urea containing 50 mM ammonium bicarbonate, and digested by 0.08 μg of Asp-N (Promega) at 37 °C for overnight. The digestion was stopped by adding 1/20 volume of 20% TFA, and digested peptides were desalted using a GL-Tip SDB (GL Science) according to the manufacturer's instructions. After vacuum concentration, the samples were dissolved in 20 μL of 2% acetonitrile (containing 0.1% TFA) and 1.5-μL aliquots were analyzed by nano-LC-MS/MS.

## Proteomic comparison between *Arabidopsis* submerged culture medium and leaf apoplastic fluid

The *Arabidopsis* leaf apoplastic fluid was collected as described previously[34] with some modifications. Briefly, 12 completely expanded leaves were vacuum-infiltrated with water. The infiltrated leaf samples were packed in a syringe and apoplastic fluid was collected by centrifugation at $410 \times g$ for 8 min. Aliquots (50 μL) of leaf apoplastic fluid were mixed with equal volume of 8 M urea containing 250 mM Tris-HCl (pH 8.5). For proteomic analysis of *Arabidopsis* submerged culture medium, proteins were extracted from 1 mL of submerged culture medium by *o*-chlorophenol extraction followed by acetone precipitation as described above, and the pellets were resuspended in 10 μL of water. 4-μL aliquots were diluted 5-fold with 8 M urea containing 250 mM Tris-HCl (pH 8.5). Protein sample solutions were reduced with 25 mM tris(2-carboxyethyl)phosphine (TCEP) at 37 °C for 15 min, alkylated using 25 mM iodoacetamide at 37 °C for 30 min in the dark with gentle shaking at 1000 rpm following the standard protocol. Proteins were then digested with 2 μg of Lys-C (FUJIFILM Wako, Japan) at 37 °C for 3 h. This Lys-C digest was diluted to a urea concentration of 2 M with 100 mM Tris-HCl (pH 8.5), followed by digestion with 2 μg of trypsin (Promega) at 37 °C overnight. The digestion was stopped by adding 1/20 volume of 20% TFA, and digested peptides were desalted using a MonoSpin C18 spin column (GL Science) according to the manufacturer's instructions. After vacuum concentration, the samples were dissolved in 20 μL of 2% acetonitrile (containing 0.1% TFA) and 7.25-μL aliquots were analyzed by nano-LC–MS/MS.

## Detection of proteolytic activity at the C-terminus of the flg22 domain using Nma-flg[47-56]-Dnp

A submerged culture medium was prepared as described above. Where applicable, the submerged culture medium was subjected to heat inactivation by incubation at 100 °C for 30 min. Nma-flg[47-56]-Dnp was added to the submerged culture medium or heat-inactivated submerged culture medium to a final concentration of 4 μM in a total volume of 100 μL per well in a 96-well white plate. The fluorescence of each well then was measured using a VICTOR Nivo Multimode Plate Reader equipped with a 340/20 nm excitation filter and a 435/20 nm emission filter; measurements were conducted every hour for 20 h.

## Native two-dimensional electrophoresis

A volume (100 mL) of submerged culture medium from WT *Arabidopsis* was concentrated 10-fold by rotary evaporation. The concentrated sample was centrifuged at $11,900 \times g$ for 5 min and the resulting supernatant was dialyzed (overnight at 4 °C) against 3 L of RO water using a Spectra/Por 7 membrane (molecular weight cut-off (MWCO), 10 kDa; Repligen). The water then was replaced with fresh water, and the sample was dialyzed for another night. The dialyzed sample was centrifuged at $11,900 \times g$ for 5 min, and the resulting

supernatant was lyophilized. The lyophilized powder was used for the native two-dimensional electrophoresis. The first dimension and gel equilibration were performed as described previously[35], with minor modifications. Briefly, separation in the first dimension, using native isoelectric focusing (IEF), was performed using an Ettan IPGphore II (Cytiva). IPG strip (Immobiline DryStrip gel pH 3-10 NL, 7 cm; Cytiva) that was rehydrated in 125 μL rehydration solution for 14 h. The lyophilized powder derived from 100 mL of spent medium from submerged culture of WT *Arabidopsis* was added to the rehydration solution before use in rehydration of the strip. IEF was run using the following program: 500 V for 30 min, 1000 V for 30 min, and 5000 V for 100 min. After the first-dimensional separation, the IPG strip was equilibrated for 15 minutes at 4 °C. The equilibrated IPG strip then was rinsed with Native PAGE Running Buffer (prepared from Native PAGE Running Buffer (20×) stock (Thermo Fisher Scientific)) and placed onto the second-dimensional gel. The second-dimensional gel was prepared by removing the wells from a commercially obtained gel (Native PAGE 4 to 16%, Bis-Tris, 1.0 mm, Mini Protein Gel, 15 well; Thermo Fisher Scientific). The anode buffer was Native PAGE Running Buffer; the cathode buffer was Native PAGE Running Buffer supplemented with 20 μg/mL Coomassie Brilliant Blue (CBB) G-250. Separation in the second dimension, by blue native PAGE, was conducted at 150 V for 90 min.

## In-gel proteolysis assay using fluorescence-quenching peptide substrate

The two-dimensional electrophoresis gel was rinsed in 25 mM 2-(N-morpholino)ethanesulfonic acid (MES) -KOH (pH 5.7) and diced using a SAINOME plate as described previously[16]. The gel was separated into 224 fragments. A solution of 40 μM Nma-flg[47-56]-Dnp in 25 mM MES-KOH (pH 5.7) then was dispensed at 60 μL/well into the SAINOME-plate, and the fluorescence of each well was measured as described above.

## In-gel protein digestion

The fluorescence-positive gel fragments, as well as control fluorescence-negative gel fragments, were collected from the respective wells of the SAINOME plate and fixed in 20% methanol and 7.5% acetic acid for 1 h. The fixed gel fragments were used for in-gel protein digestion, performed as described previously[36], with some modifications. Briefly, the fixed gel fragments were washed with MilliQ water, then with 50 mM ammonium bicarbonate in 50% acetonitrile and dried with 100% acetonitrile. The fragments were incubated in 50 mM tris(2-carboxyethyl)phosphine (TCEP) at 60 °C for 10 min. Iodoacetamide was added to a final concentration of 100 mM, and the gel fragments were incubated at room temperature in the dark for 1 h. After two washes, the gel fragments were dried with acetonitrile, and a solution of Lys-C/Trypsin (0.01 mg/mL; Promega) was added to swell the gel fragments. Then, 50 mM ammonium bicarbonate was added, and the fragments were incubated at 37 °C for 16 h. The resulting supernatants were collected, acidified to pH 2 with 20% TFA, then desalted using a Mono Spin C18 column (GL Sciences). After vacuum concentration, the samples were dissolved in 20 μL of 2% acetonitrile (containing 0.1% TFA) and 7.25-μL aliquots were analyzed by nano-LC–MS/MS.

## Mass spectrometry

Nano-LC–MS/MS analysis was performed using a system combining a Q Exactive Hybrid Quadrupole-Orbitrap Mass Spectrometer (Thermo Fisher Scientific) with a Dionex U3000 gradient pump (Thermo Fisher Scientific). For proteomic analysis, samples were loaded onto a trap column (L-column ODS [300 μm I.D. × 5 mm], CERI, Japan) and washed with 2% acetonitrile (0.1% TFA). Peptides were subsequently eluted from the trap column and separated on a nano-LC capillary column (NTCC-360/100 [100 μm I.D. × 125 mm], Nikkyo Technos, Japan) with a gradient of 5–40% acetonitrile (containing 0.5% acetic acid) over

100 min at a flow rate of 500 nL/min. For identification of in-gel digested proteins, nano-LC was operated with a gradient of 5–40% acetonitrile (containing 0.5% acetic acid) over 40 min at a flow rate of 500 nL/min. The Q Exactive mass spectrometer was operated in data-dependent acquisition mode with dynamic exclusion enabled (5 s for flagellin fragmentation analysis and identification of in-gel digested proteins; 10 s for identification of on-beads digested proteins; 20 s for proteomic analysis of *Arabidopsis* submerged culture medium and leaf apoplastic fluid) with a capillary temperature of 250 °C. Survey full-scan MS spectra (mass range 350–1800) were acquired at a resolution of 70,000 with the maximum injection time set at 60 ms. The 10 most-intense peptide ions in each survey scan with a charge state between 2+ and 4+ were selected for MS/MS fragmentation. MS/MS scans were performed by higher-energy collisional dissociation (HCD) with the normalized collision energy set to 27 (for conventional proteomics) or 35 (for flagellin fragmentation analysis). The MS/MS raw files were processed and analyzed with Proteome Discoverer 2.3 (Thermo Fisher Scientific) using the SEQUEST HT algorithm, searching against the TAIR10 *Arabidopsis* protein database, Nbe.v1 *N. benthamiana* protein database[37], or flagellin. Precursor mass tolerance and fragment mass tolerance were set at 10 ppm and 0.02 Da, respectively, and for up to 2 missed cleavages were allowed. Oxidation (M) was set as the variable modification, and carbamidomethylation (C) as the fixed modification. For the identification of apoplastic proteases, proteases with signal peptides were extracted from the resulting high-confidence master protein list using TAIR Gene Ontology (GO) annotations and SignalP 6.0[38].

For peptide cleavage assay, nano-LC–MS analysis was performed using a system combining the LTQ Orbitrap XL mass spectrometer (Thermo Fisher Scientific) with a DiNa-M splitless nano-HPLC system (KYA Technologies). A trap column (HiQ sil C18W-3, 500 μm I.D. × 1 mm; KYA Technologies) and an analytical column (MonoCap C18 Fast-Flow, 100 μm I.D. × 150 mm; GL Sciences) were used. Nano-LC was operated with a gradient of 2–50% acetonitrile (containing 0.1% formic acid) over 30 min at a flow rate of 500 nL/min. For proteolytic assay of Nma-flg[47-56]-Dnp, nano-LC was operated with a gradient of 30–80% acetonitrile (containing 0.1% formic acid) over 30 min. MS analysis was performed in positive ion mode with a capillary temperature of 160 °C. Mass spectra were obtained by scanning from $m/z$ 350 to $m/z$ 1800. For structural confirmation, MS/MS analysis was performed in positive ion mode with HCD at 35 V.

### Transient expression of proteases in *N. benthamiana*
For the transient expression of the His-tagged proteases SBT5.2-$H_6$, SBT1.7-$H_6$, and SAP2-$H_6$, individual cDNAs were obtained by RT-PCR using total RNA isolated from *Arabidopsis* leaves and gene-specific primers (Supplementary Data 4). The resulting fragments were cloned separately into the binary vector pBI121 using the In-Fusion HD cloning kit (Clontech). The resulting constructs were transformed into *Agrobacterium tumefaciens* strain C58C1 (pMP90). *Agrobacterium* strains containing plasmids encoding SBT5.2-$H_6$ or SBT1.7-$H_6$, or a strain containing a p19 silencing suppressor expression construct, was grown at 28 °C on LB agar containing 10 μg/mL rifampicin, 50 μg/mL gentamycin, and 50 μg/mL kanamycin. For each strain, following growth, an isolated colony was inoculated into 5 mL LB liquid containing 100 μM rifampicin, 100 μM gentamycin, and 100 μM kanamycin and incubated overnight at 28 °C with shaking. Bacteria were collected by centrifugation at 2000 × *g* for 5 min at room temperature, and cells from the resulting pellet were employed for agroinfiltration in *N. benthamiana* leaves, performed as described previously[39]. Five days after agroinfiltration, apoplastic fluids were collected by vacuum infiltration with 50 mM sodium phosphate buffer (pH 7.4) containing 300 mM NaCl; the resulting fluids were cleared by centrifugation at 1500 × *g* for 10 min. An aliquot (5 μL) of each resulting supernatant (the apoplastic fluid) was used for western blotting. The proteins were separated by

SDS-PAGE on a 10% acrylamide gel (REAL GEL PLATE, 10%, 16-well; Bio Craft) and transferred to a polyvinylidene difluoride (PVDF) membrane (Immobilon-P, Millipore). The membrane was blocked with Blocking One (Nacalai Tesque) at room temperature for 1 h, then incubated at room temperature for 30 min with HRP-conjugated anti-His-tag antibody (1:5000; Anti-His-Tag mAb-HRP-DirecT; MBL). The HRP signal was detected using ECL Prime Western Blotting Detection Reagent (Cytiva) and an ImageQuant LAS 4000 mini camera system (Cytiva).

### Flagellin fragmentation by SBT5.2-$H_6$ or SBT1.7-$H_6$
Aliquots (200 μL) of apoplastic fluids from agroinfiltrated leaves were diluted in buffer A (20 mM sodium phosphate buffer (pH 7.4), 500 mM NaCl, and 10 mM imidazole in MilliQ water) to a volume of 500 μL. Each sample then was loaded onto a Ni-Sepharose-packed-tip column equilibrated with 1 mL of Buffer A. The tip columns were prepared as follows: A chelating Sepharose-packed column (HiTrap chelating HP 1 mL; Cytiva) was washed with 5 mL of water, and 0.5 mL of water containing 0.1 M $NiSO_4$ was loaded onto the column. After washing with 5 mL of water, the column was equilibrated with 10 mL of buffer A. Ni-Sepharose beads were recovered from the dismantled column and suspended in 1.5 mL of Buffer A. To prepare protease-immobilized beads, a small amount of cotton was packed into a 200-μL micropipette tip, onto which 50 μL of suspended Ni-Sepharose beads were loaded. These micropipette-tip columns were loaded with the apoplast samples, then washed with 1 mL of Buffer A, and the Ni-Sepharose beads carrying immobilized proteases were collected from the micropipette-tip columns. Beads harboring immobilized SBT5.2-$H_6$, SBT1.7-$H_6$ or SAP2-$H_6$ were used for the proteolysis assay. The assays were performed in a 200-μL reaction mixture containing 25 mM MES-KOH (pH 5.7), 2 μM flagellin or 500 nM synthetic peptides, and 10 μL of protease immobilized beads. The reaction mixtures were rotated for 1 h at room temperature. Reactions were terminated by the addition of 1/10 volume of 10% TFA. For flagellin proteolysis assay, aliquots (50 μL) of the quenched reaction mixtures were desalted using a GL-Tip SDB (GL Science) according to the manufacturer's instructions. After vacuum concentration, the samples were dissolved in 20 μL of 2% acetonitrile (containing 0.1% TFA) and 3.33-μL aliquots were analyzed by nano-LC-MS/MS. For peptide proteolysis assay, aliquots (5 μL) of the quenched reaction mixtures then were analyzed by nano-LC-MS/MS.

### Protein purity of SBT5.2-$H_6$ and SBT1.7-$H_6$
The protease-immobilized Ni-Sepharose beads were suspended in 20 μl of 8 M urea containing 250 mM Tris-HCl (pH8.5), reduced with 25 mM tris(2-carboxyethyl)phosphine (TCEP) at 37 °C for 15 min and alkylated using 25 mM iodoacetamide at 37 °C for 30 min in the dark, both with shaking at 1200 rpm following the standard protocol. After dilution to a urea concentration of 2 M with 50 mM Tris-HCl (pH 8.5) followed by addition of 1 mM $CaCl_2$, proteins on beads were directly digested with 0.1 μg trypsin (Promega) with shaking at 1200 rpm at 37 °C overnight. The digestion was stopped by adding 1/20 volume of 20% TFA, and then digested peptides were desalted using a GL-Tip SDB (GL Science) according to the manufacturer's instructions. After vacuum concentration, the samples were dissolved in 20 μL of 2% acetonitrile (containing 0.1% TFA) and 7.25-μL aliquots were analyzed by nano-LC–MS/MS.

### Statistics and reproducibility
All statistical analyses were performed using Prism software (version 9; GraphPad). No statistical method was used to predetermine sample size. No data were excluded from the analysis. Samples were grown under the same conditions and randomly allocated in the growth chamber. Experimental plant material was collected randomly without any bias. Investigators were not blinded to allocation during the experiments and outcome assessment.

## Reporting summary

Further information on research design is available in the Nature Portfolio Reporting Summary linked to this article.

## Data availability

All data supporting the results of this study are available in the paper and in the Supplementary Information. All other data that supporting the results of this study are available from the corresponding author upon request. The raw MS data generated in this study have been deposited in the ProteomeXchange Consortium via the PRIDE partner repository under accession code PXD044549. Source data are provided with this paper.

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

## Acknowledgements

This research was supported by a Grant-in-Aid for Scientific Research (S) (No. 23H05477 to Y.M.), Grant-in-Aid for Transformative Research Areas (A) (No. 20H05907 to Y.M.) and the Nagoya University Interdisciplinary Frontier Fellowship (to S.M.) supported by Nagoya University and Japan Science and Technology Agency (JST) (No. JPMJFS2120).

## Author contributions

Y.M. and H.S. conceived this project and S.M., H.S., and Y.M. designed the experiments. S.M. performed all the biological experiments and S.M., H.S., and Y.M. interpreted the results. S.N. and K.K. performed the nano-LC-MS/MS analysis. M.N. and Y.T. provided support for handling *Pst* DC3000 and collecting apoplastic fluid from *Arabidopsis* leaves. S.M. and Y.M. wrote the manuscript.

## Competing interests

The authors declare no competing interests.
