## [Peer Review File · Nature Communications]

REVIEWER COMMENTS

Reviewer #1 (Remarks to the Author):

In the present study, Matsui and colleagues aim to characterize the proteolytic release of immunogenic epitopes from the flagellin protein of *Pseudomonas syringae* pv tomato DC3000. Using proteomics and in vitro biochemistry, they determine that the minimal immunogenic epitope of flagellin, flg22, can be released by extracellular proteases in *Arabidopsis thaliana*, and subsequently, through genetic and biochemical approaches, identify SBT1.7 and SBT5.2 as the major proteases responsible for immunogenic peptide release.

Major comments:

1. In general, controls demonstrating the specificity of cleavage are lacking throughout the manuscript. The authors should include an unrelated subtilase in their in vitro cleavage experiments and should additionally assess the effect of point mutants on cleavage of the fluorescent reporter peptide (i.e. alanine scanning mutagenesis) to determine specificity requirements at the substrate protein sequence level. The authors could also consider generating catalytic site mutants of SBT5.2 and SBT1.7 – this would confirm the requirement of SBT5.2 and 1.7 in bench experiments where the authors might be co-purifying endogenous proteases. Notably, both SBT1.7 and SBT5.2 were among the 4 most abundant proteases previously reported in the apoplast (Engineer et al., 2014). This may make biological sense, with copious ability to release MAMPs potentially conferring a fitness advantage. However, this also makes it more likely that these subtilases would be identified in the approach followed by the authors. This makes the determination of specificity even more important.
2. Specificity is also not clearly demonstrated in the proteomics experiments where the authors observed release of the immunogenic flg22 peptide (Figs 1b and 5c). Indeed, there is bias in the observed peptides towards the immunogenic region, but this might be a consequence of protease bias or analytical bias. Here, the authors should perform a tryptic (or other, as they see fit) digest to confirm that peptides from across the whole flagellin protein can be observed in their nanoLC-MS assays. This would provide a clear demonstration that endogenous proteases have specific behavior towards the N-terminus of flagellin.
3. There are many peptides from across the flagellin N-terminus released by plant apoplastic fluids. It is difficult to be certain, but I suspect that a number of these peptides around the immunogenic epitope would not actually trigger FLS2-dependent immune signaling or might function to antagonise receptor activation. The authors highlight that the C-terminus of flg22 seems to be a primary cleavage site, but it is likely that not all peptides with the flg22 C-terminus will elicit immune signaling because they lack portions of the flg22 N-terminus. It would also be good to test several different peptides that are released to get an idea of the diversity of biological activities – perhaps this could even be done by fractionating a digest of flagellin and testing the range of fractions for elicitor activity. In particular, it

would be good to know whether peptides extended at the N- or C-terminus of flg22 can elicit FLS2-dependent immune signaling.

4. How reproducible are the fragmentation patterns of flagellin as determined by MS? How many times have these been repeated? From my understanding, Fig. 1b and Fig. 5c (top panel) should be the same; however, there is considerable variation between these. Can the authors provide a statistical comparison? Presumably plant-derived peptides are also identified in this MS approach. Are these similar between experiment?

5. Are the authors able to observe cleavage of flagellin (full-length) with purified subtilases, as seen with the flg[47-56]? If this would not be possible to observe via MS, it could be visualised in a gel.

6. Did the authors attempt the peptidomics approach on purified flagellin which had not been mixed with the Arabidopsis apoplast? Which peptides, if any, are present in the purified flagellin without interaction with the host apoplast?

7. It is important to test whether the Nma-flg[47-56]-Dnp has any biological activity; this can be easily test in the ROS production assay.

8. The authors only tested the double sbt5.2 sbt1.7 mutant. What is the phenotype of single mutants, eg. for flagellin-induced ROS production?

9. What is the impact of the slightly delayed ROS production observed in sbt5.2 sbt1.7 regarding disease susceptibility? The authors should spray-infect WT and sbt5.2 sbt1.7 with Pst DC3000 WT, but also with a flhC mutant to test specificity of the potential phenotype to the presence of flagellin.

Minor comments:

1. Abstract, line 2: the flagellin-derived epitopes activating PTI are not specific to pathogenic bacteria, so remove "pathogenic".

2. The correct name of FLS2 is FLAGELLIN SENSING 2.

3. On Line 252, please add references after "immunogenic peptides".

4. In Figures 1b and 5c, it would be good if the authors could show a zoomed-in panel depicting peptides observed that align with part of flg22 (say, 50% alignment). This would give a clearer indication of the diversity of potentially immunogenic peptides released.

5. It would be good if the authors could include an fls2 knockout mutant in their ROS production assays to determine that another PAMP has not been co-purified.

6. Fig. 6b indicates that the sbt5.2 sbt1.7 mutant has a reduced amplitude of ROS production in response to flg22. Is this reproducible? Is this statistically significant? If so, how to explain this phenotype?

Reviewer #2 (Remarks to the Author):

Summary: Plants have a defense mechanism against pathogenic bacteria by recognizing specific epitope peptides in flagellin proteins. The proteolytic cleavage of these epitope peptides is essential for the plant's ability to detect them. However, the plant proteases responsible for this cleavage were previously unknown. In a recent study using a new identification method, two plant subtilases (SBT5.2 and SBT1.7) were identified as the specific proteases responsible for cleaving the C-terminus of the flg22 epitope in flagellin. When these proteases were deficient in Arabidopsis mutant plants, there was less cleavage of the flg22 domain, resulting in longer epitope fragments. This led to a delay in the production of defensive reactive oxygen species (ROS) in the mutant plants compared to wild-type plants when exposed to flagellin.

Overall, this manuscript offers valuable insights into the intricate mechanisms underlying plant-pathogen interactions. By investigating the role of apoplastic proteases in processing flagellin, a key pathogen-associated molecular pattern, the authors shed light on the initial stages of plant immune responses. Understanding how plant cells detect pathogen infections in the apoplastic fluid (APF), which serves as the first battleground between plants and pathogens.

Clarity and Structure:

The paper is well-structured and provides a clear introduction to the research question and objectives.

Introduction:

The introduction provides a clear context for the research, explaining the concept of pattern-triggered immunity (PTI) and the recognition of pathogen-associated molecular patterns (PAMPs or MAMPs). It also addresses the importance of proteolytic cleavage of flagellin which helps readers understand the significance of the study.

Results:

The Results section of the paper provides valuable information about the proteolytic assay of immunogenic peptides releasing from bacterial flagellin by Arabidopsis apoplastic proteases. It also discusses the fragmentation patterns of flagellin in plant apoplasts and identifies potential endo-proteases and exopeptidases involved in this process. The results are clear and explained in order which makes it easy to read and follow.

Material and Methods:

Method section is clear and detailed enough to follow the protocols.

In Figure 1-b, it is shown that multiple peptide fragments are primarily derived from the N-terminal region of flagellin, which includes the flg22 epitope peptide. Is there an explanation for why the N-terminal region of flagellin is mainly targeted by apoplastic proteases? Are there any structural differences between the N-terminal region of flagellin and other domains? It is noteworthy that flg22 is buried within the N-terminal region of the flagellin monomer, and the N-terminal region of the flagellin monomer is located in the inner part of the flagella polymer.

Previous studies have shown that flagellin is glycosylated and requires deglycosylation before being effectively digested by apoplastic proteases. How the issue of flagellin glycosylation was addressed when it was used as the starting material for digestion and ROS burst assays.

Please see here: Buscaill P, Chandrasekar B, Sanguankiatichai N, Kourelis J, Kaschani F, Thomas EL, Morimoto K, Kaiser M, Preston GM, Ichinose Y, van der Hoorn RAL. Glycosidase and glycan polymorphism control hydrolytic release of immunogenic flagellin peptides. *Science*. 2019 Apr 12;364(6436)

In the study, the authors mostly have used Arabidopsis submerged culture to investigate proteolytic processes, rather than utilizing the actual Arabidopsis apoplastic fluid (APF). While the choice of submerged culture offers several practical advantages, such as ease of handling and experimental control, it prompts questions regarding the equivalence of this system to the actual isolated APF. Did the authors perform a comparison of the protein content between Arabidopsis apoplastic fluid (APF) and the submerged culture? How confident are they that the submerged culture possesses the same protein components and functions as APF?

Reviewer #3 (Remarks to the Author):

The authors found the contribution of SBT5.2a and SBT1.7 to the site-specific endocleavage of flagellin into flg22 for the efficient PTI. The manuscript is simply written. However, this reviewer has several

concerns about the characterization and quantification of proteolytic peptides truncated by SBT5.2a and SBT1.7 in vitro and in gel. These concerns have arisen because of lack of detail especially about proteomic data analyses. This reviewer has several comments as shown below.

1. Page 1, line 24 and page 3, line 83: The authors described their method as a one-step system. This reviewer could not catch in which sense this method is a one-step system. This reviewer feels that their method consists of multiple steps.

2. Page 4, lines 143-145 and page 16, Fig. 2d: This reviewer is curious about whether the Arabidopsis submerged culture medium didn't exert any exo-proteolytic activities. Only the Nma-terminal peptides were shown in the manuscript. How was the detection of Dnp-terminal peptides? Didn't the authors detect KITK-Dnp together with TKITK-Dnp? In addition, this reviewer strongly recommends that the authors show not only precursor ions but also product ions, i.e., MS/MS profiles, since the authors evaluated proteolytic peptides based on PSM, i.e., MS/MS profiles, throughout the entire manuscript.

3. Page 5, line 176: How was the purity of hexahistidine-tagged recombinant proteins, SBT5.2a and SBT1.7, for the biochemical assays? Wasn't there contamination with endo- and/or exo-peptidases? The authors should show the profile of CBB-stained SDS-PAGE gels and, if possible, the results of shotgun proteomics to confirm the absence of any other potential endo- and/or exo-peptidases.

4. Page 5, line 177 and page 18, Fig. 4b-d: The authors still used a shorten flg[47-56] as a substrate for SBT5.2a and SBT1.7 here. Why didn't the author use a flagellin monomer as a substrate? Using the flagellin monomer seems to be beneficial to characterize the cleavage site specificity of SBT5.2a and SBT1.7. As similar above, how was the detection of TKITS and KITS peptides? This reviewer strongly recommends showing MS/MS profiles in the manuscript. Furthermore, how many replicates were done for the detection and quantification of these proteolytic peptides? Indicating only one MS1 profile for each proteolytic peptide is not enough to discuss proteolytic efficiency and cleavage site specificity. Also, how do the authors explain the gap between the results of the Arabidopsis submerged culture medium in Fig.2d and those of SBT1.7 in Fig.4c-d?

5. Page 6, lines 198-200: How many independent strains of *A. thaliana* sbt5.2 sbt1.7 double mutants were produced to compare their chemical phenotypes? In addition, complementary tests are generally required to conclude the biological significance of target genes in planta.

6. Page 6, lines 200-201 and page 19, Fig. 5a: Please indicate annealing sites of RT-PCR primers in Fig. 5a. Were forward and reverse primers designed as they both anneal upstream of T-DNA region? Can the T-DNA insertion that was found near the C-terminal region of SBT5.2 deactivate the proteolytic activity of SBT5.2 as well?

7. Page 7, lines 268-279: Are there any information about consensus sequences of cleavage sites of IDA, CRSP, and CLE40? Are these sequences similar to those of flagellin?

8. Page 12, lines 439-460, page 13, line 504, and page 14, lines 505-514 (Materials and Methods section): There are two chapters related to MS-based proteomics. Please combine them into one chapter with two paragraphs. Although this reviewer was able to find the description of how the authors operated nanoLC systems, but there was no detailed information on how Orbitrap mass spectrometers were operated. Did the authors use them under DDA control or others?

9. Page 12, lines 450-452 and 458-460 (Materials and Methods section): There are no information about how the authors performed proteomic data processing with Thermo Proteome Discoverer 2.3/2.4. Especially, the authors should describe the detailed parameters of PSM searches by Thermo Sequest HT. For example, how was the mass tolerance of precursor ions and fragment ions?

Reviewer #1 (Remarks to the Author):

In the present study, Matsui and colleagues aim to characterize the proteolytic release of immunogenic epitopes from the flagellin protein of *Pseudomonas syringae* pv tomato DC3000. Using proteomics and in vitro biochemistry, they determine that the minimal immunogenic epitope of flagellin, flg22, can be released by extracellular proteases in *Arabidopsis thaliana*, and subsequently, through genetic and biochemical approaches, identify SBT1.7 and SBT5.2 as the major proteases responsible for immunogenic peptide release.

Thank you for taking the time to evaluate our manuscript.

Major comments:

1. In general, controls demonstrating the specificity of cleavage are lacking throughout the manuscript. The authors should include an unrelated subtilase in their in vitro cleavage experiments and should additionally assess the effect of point mutants on cleavage of the fluorescent reporter peptide (i.e. alanine scanning mutagenesis) to determine specificity requirements at the substrate protein sequence level. The authors could also consider generating catalytic site mutants of SBT5.2 and SBT1.7 – this would confirm the requirement of SBT5.2 and 1.7 in benth experiments where the authors might be co-purifying endogenous proteases. Notably, both SBT1.7 and SBT5.2 were among the 4 most abundant proteases previously reported in the apoplast (Engineer et al., 2014). This may make biological sense, with copious ability to release MAMPs potentially conferring a fitness advantage. However, this also makes it more likely that these subtilases would be identified in the approach followed by the authors. This makes the determination of specificity even more important.

Attempts to express 4 unrelated SBTs (SBT1.8, SBT3.13, SBT5.6, SBT4.13) and a catalytic site mutant of SBT5.2 in tobacco leaves were unsuccessful, with low expression or no expression at all, under conditions where SBT5.2 (as a positive control) was efficiently expressed. Instead, SAP2 (AT1G03220), an unrelated protease that was non-specifically detected in gel fractions (Supplementary Dataset 2), was successfully expressed and used in the flagellin cleavage experiment. When flagellin was digested by SAP2, the Ala⁵¹-Thr⁵² site was not cleaved, confirming the substrate specificity of SBT5.2 and SBT1.7 (Supplementary Fig. 3b, c). An alanine-scanning experiment demonstrated that residues in the P2 position (Ile⁵⁰) affect the cleavage efficiency by SBT5.2-H₆ at this site (Supplementary Fig. 3a).

Additionally, we confirmed that flagellin does not fragment when incubated with Ni-Sepharose beads exposed to control apoplastic fluid collected from *N. benthamiana* plants expressing the empty vector (Supplementary Fig. 3d). We further confirmed that immobilized *Arabidopsis* SBT5.2-H₆ and SBT1.7-H₆ contain few or no detectable contamination by *N. benthamiana* proteases, thus excluding the possibility that cleavage of flagellin reflects co-purified *N. benthamiana* proteases (Supplementary Dataset 3).

2. Specificity is also not clearly demonstrated in the proteomics experiments where the authors observed release of the immunogenic flg22 peptide (Figs 1b and 5c). Indeed, there is bias in the observed peptides towards the immunogenic region, but this might be a consequence of protease bias or analytical bias. Here, the authors should perform a tryptic (or other, as they see fit) digest to confirm that peptides from across the whole flagellin protein can be observed in their nanoLC-MS assays. This would provide a clear demonstration that endogenous proteases have specific behavior towards the N-terminus of flagellin.

When flagellin was cleaved by the endoprotease Asp-N, which cleaves peptide bonds N-terminal to Asp residues, peptide fragments were detected from across virtually the entire length of the protein, except for the Ser¹⁴³-Ser²⁰¹ domain to which carbohydrate chains are known to be attached (Supplementary Fig. 2a, b). These results indicated the presence of apoplastic proteases that predominantly cleave the flagellin N-terminal region, including those that release the flg22 epitope.

3. There are many peptides from across the flagellin N-terminus released by plant apoplastic fluids. It is difficult to be certain, but I suspect that a number of these peptides around the immunogenic epitope would not actually trigger FLS2-dependent immune signaling or might function to antagonise receptor activation. The authors highlight that the C-terminus of flg22 seems to be a primary cleavage site, but it is likely that not all peptides with the flg22 C-terminus will elicit immune signaling because they lack portions of the flg22 N-terminus. It would also be good to test several different peptides that are released to get an idea of the diversity of biological activities – perhaps this could even be done by fractionating a digest of flagellin and testing the range of fractions for elicitor activity. In particular, it would be good to know whether peptides extended at the N- or C-terminus of flg22 can elicit FLS2-dependent immune signaling.

The activity of flg22 variants lacking the N or C terminus residues has already been described in detail (J. Biol. Chem. 276, 45669-45676 (2001)). Activity of flg15, which lacks seven N-terminal residues, is about 1/100 of flg22. Additionally, flg13, which lacks 9 N-terminal residues, is completely inactive. On the other hand, flg22 Δ 2, which lacks two C-terminal residues, is known to function as a weak antagonist, but at the levels detected in Fig. 1c, its effect on the defense response would be negligible.

However, since the activity of flg22 peptides with extended sequences has not been investigated, we used synthetic peptides to examine their effect on ROS production. Since we are interested in the C-terminal cleavage of flg22 domain, we synthesized flg22C6 or flg22C12, a peptide consisting of a 6- or 12-residue extension of the C-terminus of flg22. Treatment of WT leaf disks with flg22C6 resulted in delayed ROS production (Supplementary Fig. 1c, d). flg22C12 also caused a similar delay in ROS production (data not shown in the manuscript). These observations again suggest that the C-terminal cleavage of the flg22 epitope is important for activation of the defense response.

4. How reproducible are the fragmentation patterns of flagellin as determined by MS? How many times have these been repeated? From my understanding, Fig. 1b and Fig. 5c (top panel) should be the same; however, there is considerable variation between these. Can the authors provide a statistical comparison? Presumably plant-derived peptides are also identified in this MS approach. Are these similar between experiment?

In the revised manuscript, we provided statistical data showing frequency at which each amino acid residue of flagellin is the C-terminal end or N-terminal end of the peptide fragments following digestion by proteases (Figs. 1d, 4b, 4d, 5c, 5e, S2b, S5b-d). Each bar represents the mean value and error obtained from three or more experiments. Even though the total PSM values of peptide fragments obtained by mass spectrometry differ from experiment to experiment, the frequency of cleavage at each site is similar, so we believe these figures provide a good indicator for comparing cleavage patterns.

Fragments of various *Arabidopsis* proteins are also detected in the submerged culture medium, but their profiles do not always seem to be the same from experiment to experiment,

perhaps because they are affected by variations in expression levels of both proteases and substrates.

5. Are the authors able to observe cleavage of flagellin (full-length) with purified subtilases, as seen with the flg[47-56]? If this would not be possible to observe via MS, it could be visualised in a gel.

The same advice was given by Reviewer #3. We found that the cleavage assay using the full-length flagellin provided much more information than the cleavage assay using the short peptide, so in the revised manuscript we present the results of the cleavage assay using the full-length flagellin.

Incubation of flagellin protein with SBT5.2-H₆ or SBT1.7-H₆ for 1 h followed by nano-LC-MS/MS analysis confirmed that both SBT5.2-H₆ and SBT1.7-H₆ cleave precisely at the C-terminus of the flg22 domain (Ala⁵¹-Thr⁵²) (Fig. 4a-d). In addition, SBT5.2-H₆ and SBT1.7-H₆ provided proteolytic cleavage of flagellin at Asn³⁹-Ser⁴⁰, a site located in the central region of the flg22 domain (Fig. 4a-d). Since the flg22 epitope loses its immunogenic activity when cleaved at Asn³⁹-Ser⁴⁰, the balance of the cleavage efficiency between Asn³⁹-Ser⁴⁰ and Ala⁵¹-Thr⁵² is critical for liberation of active immunogenic peptides. We also detected minor but multiple cleavage sites within the N-terminal region of flagellin (Fig. 4a-d). These results indicated that SBT5.2 and SBT1.7 have preferential cleavage target sequences, although selection of the cleavage site is not always stringent.

6. Did the authors attempt the peptidomics approach on purified flagellin which had not been mixed with the Arabidopsis apoplast? Which peptides, if any, are present in the purified flagellin without interaction with the host apoplast?

We confirmed that few peptide fragments were detected in mock-treated flagellin (Supplementary Fig. 1b).

7. It is important to test whether the Nma-flg[47-56]-Dnp has any biological activity; this can be easily test in the ROS production assay.

Since flg[47-56] lacks the N-terminal 17 residues of flg22, Nma-flg[47-56]-Dnp has no biological activity. Nma-flg[47-56]-Dnp is designed to detect cleavage activity only.

8. The authors only tested the double *sbt5.2 sbt1.7* mutant. What is the phenotype of single mutants, eg. for flagellin-induced ROS production?

We evaluated the activity of single mutants in a flagellin fragmentation assay. We observed decreased proteolytic activity at the C-terminus of the flg22 domain in submerged culture medium from the *sbt5.2* single mutant, compared to that of WT (Supplementary Fig. 5b). In contrast, the flagellin digestion pattern observed in submerged culture medium from the *sbt1.7* single mutant was similar to that obtained from the WT (Supplementary Fig. 5c). These results indicated that SBT5.2 is the major apoplastic protease responsible for the cleavage of flagellin at the C-terminus of the flg22 domain *in planta*, and that SBT1.7 also functions redundantly in the generation of this immunogenic peptide.

9. What is the impact of the slightly delayed ROS production observed in *sbt5.2 sbt1.7* regarding disease susceptibility? The authors should spray-infect WT and *sbt5.2 sbt1.7* with

Pst DC3000 WT, but also with a *fliC* mutant to test specificity of the potential phenotype to the presence of flagellin.

Prior to the initial submission of this paper, we examined for more than a year various methods, including spray-infect, to find differences in susceptibility to Pst DC3000 between the WT and the *sbt5.2 sbt1.7* mutant. However, no significant differences were found in lesion formation or bacterial growth activity. Since the *sbt5.2 sbt1.7* mutant still have a slight residual activity to liberate immunogenic peptides, activation of FLS2 appears to be inevitable, albeit delayed compared to WT. Identification of the enzyme responsible for the residual activity to cleave the C-terminal side of the *flg22* domain and the enzyme responsible for the N-terminal side will be our next research target.

Minor comments:

1. Abstract, line 2: the flagellin-derived epitopes activating PTI are not specific to pathogenic bacteria, so remove “pathogenic”.

We have removed the word "pathogenic".

2. The correct name of FLS2 is FLAGELLIN SENSING 2.

We have corrected the relevant text.

3. On Line 252, please add references after “immunogenic peptides”.

We have added a reference to this sentence.

4. In Figures 1b and 5c, it would be good if the authors could show a zoomed-in panel depicting peptides observed that align with part of *flg22* (say, 50% alignment). This would give a clearer indication of the diversity of potentially immunogenic peptides released.

In the revised manuscript, the figures have been enlarged in width for better readability.

5. It would be good if the authors could include an *fls2* knockout mutant in their ROS production assays to determine that another PAMP has not been co-purified.

We also included *fls2* knockout mutants in the ROS production assay (Supplementary Fig. 1a).

6. Fig. 6b indicates that the *sbt5.2 sbt1.7* mutant has a reduced amplitude of ROS production in response to *flg22*. Is this reproducible? Is this statistically significant? If so, how to explain this phenotype?

Although the response of the *sbt5.2 sbt1.7* mutant may appear to be smaller than that of the WT in this figure, there is no statistically significant difference.

Reviewer #2 (Remarks to the Author):

Summary: Plants have a defense mechanism against pathogenic bacteria by recognizing specific epitope peptides in flagellin proteins. The proteolytic cleavage of these epitope peptides is essential for the plant's ability to detect them. However, the plant proteases responsible for this cleavage were previously unknown. In a recent study using a new identification method, two plant subtilases (SBT5.2 and SBT1.7) were identified as the specific proteases responsible for cleaving the C-terminus of the flg22 epitope in flagellin. When these proteases were deficient in Arabidopsis mutant plants, there was less cleavage of the flg22 domain, resulting in longer epitope fragments. This led to a delay in the production of defensive reactive oxygen species (ROS) in the mutant plants compared to wild-type plants when exposed to flagellin.

Overall, this manuscript offers valuable insights into the intricate mechanisms underlying plant-pathogen interactions. By investigating the role of apoplastic proteases in processing flagellin, a key pathogen-associated molecular pattern, the authors shed light on the initial stages of plant immune responses. Understanding how plant cells detect pathogen infections in the apoplastic fluid (APF), which serves as the first battleground between plants and pathogens.

Clarity and Structure:

The paper is well-structured and provides a clear introduction to the research question and objectives.

Introduction:

The introduction provides a clear context for the research, explaining the concept of pattern-triggered immunity (PTI) and the recognition of pathogen-associated molecular patterns (PAMPs or MAMPs). It also addresses the importance of proteolytic cleavage of flagellin which helps readers understand the significance of the study.

Results:

The Results section of the paper provides valuable information about the proteolytic assay of immunogenic peptides releasing from bacterial flagellin by Arabidopsis apoplastic proteases. It also discusses the fragmentation patterns of flagellin in plant apoplasts and identifies potential endo-proteases and exopeptidases involved in this process. The results are clear and explained in order which makes it easy to read and follow.

Material and Methods:

Method section is clear and detailed enough to follow the protocols.

Thank you very much for your high evaluation of our manuscript.

Figure 1-b, it is shown that multiple peptide fragments are primarily derived from the N-terminal region of flagellin, which includes the flg22 epitope peptide. Is there an explanation for why the N-terminal region of flagellin is mainly targeted by apoplastic proteases? Are there any structural differences between the N-terminal region of flagellin and other domains? It is noteworthy that flg22 is buried within the N-terminal region of the flagellin monomer, and the N-terminal region of the flagellin monomer is located in the inner part of the flagella polymer.

Reviewer #1 asked the same question. When flagellin was cleaved by the endoprotease Asp-N, which cleaves peptide bonds N-terminal to Asp residues, peptide fragments were detected from across virtually the entire length of the protein, except for the Ser¹⁴³-Ser²⁰¹ domain to which carbohydrate chains are known to be attached (Supplementary Fig. 2a, b). Therefore, the reason why the N-terminal side of flagellin is preferentially cleaved is not due to structural factors of flagellin, but to sequence specificity on the enzyme side.

Previous studies have shown that flagellin is glycosylated and requires deglycosylation before being effectively digested by apoplastic proteases. How the issue of flagellin glycosylation was addressed when it was used as the starting material for digestion and ROS burst assays. Please see here: Buscaill P, Chandrasekar B, Sanguankiatichai N, Kourelis J, Kaschani F, Thomas EL, Morimoto K, Kaiser M, Preston GM, Ichinose Y, van der Hoorn RAL. Glycosidase and glycan polymorphism control hydrolytic release of immunogenic flagellin peptides. *Science*. 2019 Apr 12;364(6436)

Because At2g28470, the *Arabidopsis* homolog of tobacco BGAL8, is also detected in submerged culture medium (Supplementary Dataset 1), it may facilitate flg22 cleavage. However, since purified SBT5.2-His₆ alone can cleave flagellin, it appears that flagellin is cleaved to some extent even in the absence of BGAL8.

In the study, the authors mostly have used *Arabidopsis* submerged culture to investigate proteolytic processes, rather than utilizing the actual *Arabidopsis* apoplastic fluid (APF). While the choice of submerged culture offers several practical advantages, such as ease of handling and experimental control, it prompts questions regarding the equivalence of this system to the actual isolated APF. Did the authors perform a comparison of the protein content between *Arabidopsis* apoplastic fluid (APF) and the submerged culture? How confident are they that the submerged culture possesses the same protein components and functions as APF?

We compared secreted proteins in submerged culture medium and actual apoplastic fluid by proteomics. Of the secreted proteins detected in the actual leaf apoplast, 75% also were detected in the submerged culture medium (Supplementary Dataset 1).

Reviewer #3 (Remarks to the Author):

The authors found the contribution of SBT5.2a and SBT1.7 to the site-specific endocleavage of flagellin into flg22 for the efficient PTI. The manuscript is simply written. However, this reviewer has several concerns about the characterization and quantification of proteolytic peptides truncated by SBT5.2a and SBT1.7 in vitro and in gel. These concerns have arisen because of lack of detail especially about proteomic data analyses. This reviewer has several comments as shown below.

Thank you very much for your comments which help us to improve the manuscript.

1. Page 1, line 24 and page 3, line 83: The authors described their method as a one-step system. This reviewer could not catch in which sense this method is a one-step system. This reviewer feels that their method consists of multiple steps.

“One-step system” is rephrased as “efficient system”.

2. Page 4, lines 143-145 and page 16, Fig. 2d: This reviewer is curious about whether the *Arabidopsis* submerged culture medium didn't exert any exo-proteolytic activities. Only the Nma-terminal peptides were shown in the manuscript. How was the detection of Dnp-terminal peptides? Didn't the authors detect KITK-Dnp together with TKITK-Dnp? In addition, this reviewer strongly recommends that the authors show not only precursor ions but also product ions, i.e., MS/MS profiles, since the authors evaluated proteolytic peptides based on PSM, i.e., MS/MS profiles, throughout the entire manuscript.

TKITK-Dnp peptide was also detected, but was difficult to quantify because it exhibited significant tailing. MS/MS spectrum of the N-terminal Nma-GLQIA peptide was shown in Fig. 2e. MS/MS spectra are also shown for other peptide-based assay results (Supplementary Fig. 4b, d, f).

3. Page 5, line 176: How was the purity of hexahistidine-tagged recombinant proteins, SBT5.2a and SBT1.7, for the biochemical assays? Wasn't there contamination with endo- and/or exo-peptidases? The authors should show the profile of CBB-stained SDS-PAGE gels and, if possible, the results of shotgun proteomics to confirm the absence of any other potential endo- and/or exo-peptidases.

We confirmed by shotgun proteomics that immobilized *Arabidopsis* SBT5.2-H₆ and SBT1.7-H₆ contain few or no detectable contamination by *N. benthamiana* proteases (Supplementary Dataset 3). We also confirmed that flagellin does not fragment when incubated with unrelated protease SAP2-H₆ (Supplementary Fig. 3b, c) or control beads exposed to apoplastic fluid collected from *N. benthamiana* plants expressing the empty vector (Supplementary Fig. 3d), thus excluding the possibility that cleavage of the flagellin reflects co-purified *N. benthamiana* proteases.

4. Page 5, line 177 and page 18, Fig. 4b-d: The authors still used a shorten flg[47-56] as a substrate for SBT5.2a and SBT1.7 here. Why didn't the author use a flagellin monomer as a substrate? Using the flagellin monomer seems to be beneficial to characterize the cleavage site specificity of SBT5.2a and SBT1.7. As similar above, how was the detection of TKITS and KITS peptides? This reviewer strongly recommends showing MS/MS profiles in the manuscript. Furthermore, how many replicates were done for the detection and quantification

of these proteolytic peptides? Indicating only one MS1 profile for each proteolytic peptide is not enough to discuss proteolytic efficiency and cleavage site specificity. Also, how do the authors explain the gap between the results of the Arabidopsis submerged culture medium in Fig.2d and those of SBT1.7 in Fig.4c-d?

The same advice was given by Reviewer #1. We found that the cleavage assay using the full-length flagellin provided much more information than the cleavage assay using the short peptide, so in the revised manuscript we present the results of the cleavage assay using the full-length flagellin instead of synthetic short peptides.

Incubation of flagellin protein with SBT5.2-H₆ or SBT1.7-H₆ for 1 h followed by nano-LC-MS/MS analysis confirmed that both SBT5.2-H₆ and SBT1.7-H₆ cleave precisely at the C-terminus of the flg22 domain (Ala⁵¹-Thr⁵²) (Fig. 4a-d). In addition, SBT5.2-H₆ and SBT1.7-H₆ provided proteolytic cleavage of flagellin at Asn³⁹-Ser⁴⁰, a site located in the central region of the flg22 domain (Fig. 4a-d). Since the flg22 epitope loses its immunogenic activity when cleaved at Asn³⁹-Ser⁴⁰, the balance of the cleavage efficiency between Asn³⁹-Ser⁴⁰ and Ala⁵¹-Thr⁵² is critical for liberation of active immunogenic peptides. We also detected minor but multiple cleavage sites within the N-terminal region of flagellin (Fig. 4a-d). These results indicated that SBT5.2 and SBT1.7 have preferential cleavage target sequences, but selection of the cleavage site is not always stringent.

In the revised manuscript, we also added new figures showing frequency at which each amino acid residue of flagellin is the C-terminal end or N-terminal end of the peptide fragments following digestion by proteases (Figs. 1d, 4b, 4d, 5c, 5e, S2b, S5b-d). Each bar represents the mean value and error obtained from three or more experiments. Although the total PSM values of peptide fragments obtained by mass spectrometry differ from experiment to experiment, the frequency of cleavage at each site is similar, so we believe that these figures are a good indicator for comparing cleavage patterns.

5. Page 6, lines 198-200: How many independent strains of *A. thaliana* *sbt5.2 sbt1.7* double mutants were produced to compare their chemical phenotypes? In addition, complementary tests are generally required to conclude the biological significance of target genes in planta.

Although only one *sbt5.2 sbt1.7* double mutant line was produced, the results of the complementation experiment with *SBT5.2* gene were newly added. Submerged culture medium from a complementation line, in which the *SBT5.2* gene was introduced into the *sbt5.2 sbt1.7* double mutant, showed proteolytic cleavage patterns comparable to those observed with the submerged culture medium from *sbt1.7* single-mutant or WT plants (Supplementary Fig. 5d, e).

6. Page 6, lines 200-201 and page 19, Fig. 5a: Please indicate annealing sites of RT-PCR primers in Fig. 5a. Were forward and reverse primers designed as they both anneal upstream of T-DNA region? Can the T-DNA insertion that was found near the C-terminal region of SBT5.2 deactivate the proteolytic activity of SBT5.2 as well?

The figure showing T-DNA insertion sites and the RT-PCR figure were separated into two independent figures due to space limitation in the revised manuscript. Therefore, the annealing sites of RT-PCR primers are not shown in the figure, but in the Supplementary Table 1.

Indeed, the T-DNA insertion site of the *sbt5.2* mutant is located near the C-terminus, but the loss of enzyme activity was confirmed by complementation experiments (Supplementary Fig. 5d, e).

7. Page 7, lines 268-279: Are there any information about consensus sequences of cleavage sites of IDA, CRSP, and CLE40? Are these sequences similar to those of flagellin?

It is interesting that SBT5.2 and SBT1.7 efficiently cleave at least two sites in flagellin, Ala⁵¹-Thr⁵² and Asn³⁹-Ser⁴⁰, and in both sites the residue in the P2 position is Ile. However, no Ile is found in the P2 position of the cleavage site in IDA, EPF2, and CLE40, so there does not appear to be a definitive consensus sequence.

8. Page 12, lines 439-460, page 13, line 504, and page 14, lines 505-514 (Materials and Methods section): There are two chapters related to MS-based proteomics. Please combine them into one chapter with two paragraphs. Although this reviewer was able to find the description of how the authors operated nanoLC systems, but there was no detailed information on how Orbitrap mass spectrometers were operated. Did the authors use them under DDA control or others?

The two independent chapters related to the operating conditions of the mass spectrometer have been arranged to be two consecutive chapters, rather than combined into one chapter. The first chapter describes shotgun proteomics by Q Exactive, and the second chapter describes the operation of peptide cleavage assay by LTQ Orbitrap. Operating parameters of the mass spectrometer are also described in detail.

9. Page 12, lines 450-452 and 458-460 (Materials and Methods section): There are no information about how the authors performed proteomic data processing with Thermo Proteome Discoverer. Especially, the authors should describe the detailed parameters of PSM searches by Thermo Sequest HT. For example, how was the mass tolerance of precursor ions and fragment ions?

A detailed description of data processing with Thermo Proteome Discoverer has been added in the revised manuscript. Precursor mass tolerance and fragment mass tolerance were set at 10 ppm and 0.02 Da, respectively, and for up to 2 missed cleavages were allowed.

REVIEWERS' COMMENTS

Reviewer #1 (Remarks to the Author):

We wish to thank the authors for their thorough revision of the manuscript. All points that we raised in our original review have been adequately addressed.

I find it very interesting that the same proteases cleave at the C-terminus and between Asn39-Ser40. This could be a potential mechanism by which the bacteria try to mask flagellin immunogenicity. The authors may wish to discuss this in more detail, or even look at the presence of these P2 Ile residues in diverse flagellin sequences.

Reviewer #3 (Remarks to the Author):

The authors performed additional experiments to further characterize the functionality of SBT5.2 and SBT1.7. They have also added more details, including those on proteomic analyses to the revised manuscript. The revised manuscript is more informative; it has been surely improved. This reviewer has only several minor comments as shown below.

1. TKITK-Dnp peptide was also detected, but was difficult to quantify because it exhibited significant tailing. MS/MS spectrum of the N-terminal Nma-GLQIA peptide was shown in Fig. 2e. MS/MS spectra are also shown for other peptide-based assay results (Supplementary Fig. 4b, d, f).

>Please attach detected, representative profiles of precursor ions and product ions of TKITK-Dnp to Supplementary information for better qualitative characterization of site-specific proteolysis.

2. Incubation of flagellin protein with SBT5.2-H6 or SBT1.7-H6 for 1 h followed by nano-LC-MS/MS analysis confirmed that both SBT5.2-H6 and SBT1.7-H6 cleave precisely at the C-terminus of the flg22 domain (Ala51-Thr52) (Fig. 4a-d). In addition, SBT5.2-H6 and SBT1.7-H6 provided proteolytic cleavage of flagellin at Asn39-Ser40, a site located in the central region of the flg22 domain (Fig. 4a-d). Since the flg22 epitope loses its immunogenic activity when cleaved at Asn39-Ser40, the balance of the cleavage efficiency between Asn39-Ser40 and Ala51-Thr52 is critical for liberation of active immunogenic peptides. We also detected minor but multiple cleavage sites within the N-terminal region of flagellin

(Fig. 4a-d). These results indicated that SBT5.2 and SBT1.7 have preferential cleavage target sequences, but selection of the cleavage site is not always stringent.

>Please attach representative profiles of precursor ions and product ions to Supplementary information. Also, this reviewer strongly recommends moving the results of alanine-scanning substitution (that the authors newly added to Supplementary Fig. 3a) to the main text.

3. The figure showing T-DNA insertion sites and the RT-PCR figure were separated into two independent figures due to space limitation in the revised manuscript. Therefore, the annealing sites of RT-PCR primers are not shown in the figure, but in the Supplementary Table 1.

>The results of RT-PCRs for *A. thaliana* sbt5.2 sbt1.7 double mutants were moved to an inappropriate position (Supplementary Fig. 5a). The authors should display them in the main text together with the illustration of T-DNA insertion sites (Fig. 5a). Furthermore, the authors should visualize the annealing sites of RT-PCR primers in Fig. 5a; otherwise, readers and I can not readily reach them, and it is not easy to judge the validity of the experiment and the samples used in this study. Also, this reviewer can not find the results of RT-PCRs for the other three mutants in Supplementary Fig. 5.

4. The two independent chapters related to the operating conditions of the mass spectrometer have been arranged to be two consecutive chapters, rather than combined into one chapter. The first chapter describes shotgun proteomics by Q Exactive, and the second chapter describes the operation of peptide cleavage assay by LTQ Orbitrap.

>The titles of the two chapters are very similar; it is unclear and not easy to distinguish. If the authors still want to split a description on nano-LC-MS/MS analysis in two, they should reconsider these titles for readers. For example, the authors could title them "shotgun proteomics" and "peptide cleavage assay".

Reviewer #1 (Remarks to the Author):

1. I find it very interesting that the same proteases cleave at the C-terminus and between Asn³⁹-Ser⁴⁰. This could be a potential mechanism by which the bacteria try to mask flagellin immunogenicity. The authors may wish to discuss this in more detail, or even look at the presence of these P2 Ile residues in diverse flagellin sequences.

The following text was added to the discussion. “An Ile residue required for SBT5.2 recognition is also found at the P2 position of the Asn³⁹-Ser⁴⁰ cleavage site, and this Ile-Asn-Ser sequence is conserved in the flg22 domain of flagellin proteins from different bacterial species.”

Reviewer #3 (Remarks to the Author):

1. Please attach detected, representative profiles of precursor ions and product ions of TKITK-Dnp to Supplementary information for better qualitative characterization of site-specific proteolysis.

In accordance with the comments, we have included the MS/MS spectral data of TKITK-Dnp in Supplementary Fig. 3a, b.

2. Incubation of flagellin protein with SBT5.2-H₆ or SBT1.7-H₆ for 1 h followed by nano-LC-MS/MS analysis confirmed that both SBT5.2-H₆ and SBT1.7-H₆ cleave precisely at the C-terminus of the flg22 domain (Ala⁵¹-Thr⁵²) (Fig. 4a-d). Please attach representative profiles of precursor ions and product ions to Supplementary information. Also, this reviewer strongly recommends moving the results of alanine-scanning substitution (that the authors newly added to Supplementary Fig. 3a) to the main text.

Although it is not common to present individual MS/MS data in shotgun proteomics, the MS/MS spectrum of the flg22 epitope peptide cleaved from flagellin by SBT5.2-H₆ is shown in Fig. 4e as a representative example. We also moved the results of alanine-scanning experiments to the main text (Fig. 4f).

3. The results of RT-PCRs for *A. thaliana* *sbt5.2 sbt1.7* double mutants were moved to an inappropriate position (Supplementary Fig. 5a). The authors should display them in the main text together with the illustration of T-DNA insertion sites (Fig. 5a). Furthermore, the authors should visualize the annealing sites of RT-PCR primers in Fig. 5a; otherwise, readers and I can not readily reach them, and it is not easy to judge the validity of the experiment and the samples used in this study. Also, this reviewer can not find the results of RT-PCRs for the other three mutants in Supplementary Fig. 5.

We have returned the RT-PCR results of the *sbt5.2 sbt1.7* double mutant to the main text, together with the illustration of T-DNA insertion sites (Fig. 5a, b). We have also marked the annealing sites of the RT-PCR primers with arrows (Fig. 5a). The RT-PCR results for the other three mutants are shown in Supplementary Fig. 6a.

4. If the authors still want to split a description on nano-LC-MS/MS analysis in two, they should reconsider these titles for readers. For example, the authors could title them “shotgun proteomics” and “peptide cleavage assay”.

In accordance with previous comments, we have combined them into one chapter.